# Long non-coding subgenomic flavivirus RNAs have extended 3D structures and are flexible in solution

Yupeng Zhang[1],[†], Yikan Zhang[1],[†], Zhong-Yu Liu[2],[3],[4],[†], Meng-Li Cheng[2],[†], Junfeng Ma[1], Yan Wang[1], Cheng-Feng Qin[2],[*] & Xianyang Fang[1],[**] (iD)

## Abstract

**Most mosquito-borne flaviviruses, including Zika virus (ZIKV), Dengue virus (DENV), and West Nile virus (WNV), produce long non-coding subgenomic RNAs (sfRNAs) in infected cells that link to pathogenicity and immune evasion. Until now, the structural characterization of these lncRNAs remains limited. Here, we studied the 3D structures of individual and combined subdomains of sfRNAs, and visualized the accessible 3D conformational spaces of complete sfRNAs from DENV2, ZIKV, and WNV by small angle X-ray scattering (SAXS) and computational modeling. The individual xrRNA1s and xrRNA2s adopt similar structures in solution as the crystal structure of ZIKV xrRNA1, and all xrRNA1-2s form compact structures with reduced flexibility. While the DB12 of DENV2 is extended, the DB12s of ZIKV and WNV are compact due to the formation of intertwined double pseudoknots. All 3′ stem-loops (3′SLs) share similar rod-like structures. Complete sfRNAs are extended and sample a large conformational space in solution. Our work not only provides structural insight into the function of flavivirus sfRNAs, but also highlights strategies of visualizing other lncRNAs in solution by SAXS and computational methods.**

**Keywords** 3D structure; computational modeling; long non-coding RNA; small angle X-ray scattering; subgenomic flavivirus RNA

**Subject Categories** Microbiology, Virology & Host Pathogen Interaction; Structural Biology

## Introduction

Long non-coding RNAs (lncRNAs) are an expanding group of cellular transcripts that range from 200 nt to over 100 kb in length and possess no protein-coding potential [1]. Like their host cells, many if not all viruses can make their own lncRNAs with multiple biological functions, including the regulation of viral replication, viral persistence, host immune evasion, and pathogenesis [2,3]. These diverse roles of lncRNAs are dictated by their propensities to form stable and complex secondary and higher-order structures, as evidenced by the recent research on Kaposi's sarcoma-associated herpesvirus (KSHV) polyadenylated nuclear (PAN) RNA which a viral lncRNA consists of complex structures [4].

Mosquito-borne flaviviruses, such as Zika virus (ZIKV), Dengue virus (DENV), and West Nile virus (WNV), are important human pathogens that cause several million deaths and hundreds of millions of cases each year [5–7]. Due to the lack of effective antiviral drugs and vaccines against these viruses, they pose a significant threat to human health and are serious concerns in many parts of the world [8]. Flaviviruses are enveloped RNA viruses with single-stranded, positive sense genomic RNA (gRNA) of approximately 10–11 kb in length which consists of a single open reading frame (ORF) flanked by highly structured 5′ and 3′ untranslated regions (UTR). The ORF encodes a polyprotein which is cotranslationally and/or posttranslationally processed into three structural proteins (capsid, pre-membrane/membrane, and envelope) and seven non-structural proteins (NS1, NS2A, NS2B, NS3, NS4A, NS4B, and NS5) [9]. The 5′ UTR, 3′ UTR, and capsid coding sequences are highly structured and contain many cis-acting elements which are involved in gRNA replication, translation, and perhaps encapsidation [9–11].

It has been reported that an incomplete degradation of the flavivirus gRNA by stalling of the cellular 5′→3′ exonuclease Xrn1 near the beginning of the 3′ UTR is responsible for the production of an abundance of long non-coding subgenomic flavivirus RNAs (sfRNA) ranging from 300 to 500 nt in length during infection [9,12]. In DENV2, the 3′ UTR is structurally divided into five independently folded subdomains: SLI, SLII, DB1, DB2, and the essential terminal structure 3′ SL (Fig 1A) [13]. Intriguingly, SLI and SLII, DB1 and DB2, respectively, are conserved structure duplications with sequences involved in pseudoknot formation (PK1, PK2, PK3, PK4, respectively; Fig 1A). Recent report showed that halting RNA degradation just upstream of each of the stem-loops (SLI and SLII) and the dumbbell structures (DB1 and DB2) results in the formation of

1   Beijing Advanced Innovation Center for Structural Biology, School of Life Sciences, Tsinghua University, Beijing, China
2   State Key Laboratory of Pathogen and Biosecurity, Beijing Institute of Microbiology and Epidemiology, Beijing, China
3   Guangzhou Eighth People's Hospital, Guangzhou Medical University, Guangzhou, China
4   School of Medicine (Shenzhen), Sun Yat-sen University, Guangzhou, China
    *Corresponding author. Tel: +86 10 66948604; E-mail: qincf@bmi.ac.cn
    **Corresponding author. Tel: +86 10 62771071; E-mail: fangxy@tsinghua.edu.cn
    †These authors contributed equally to this work

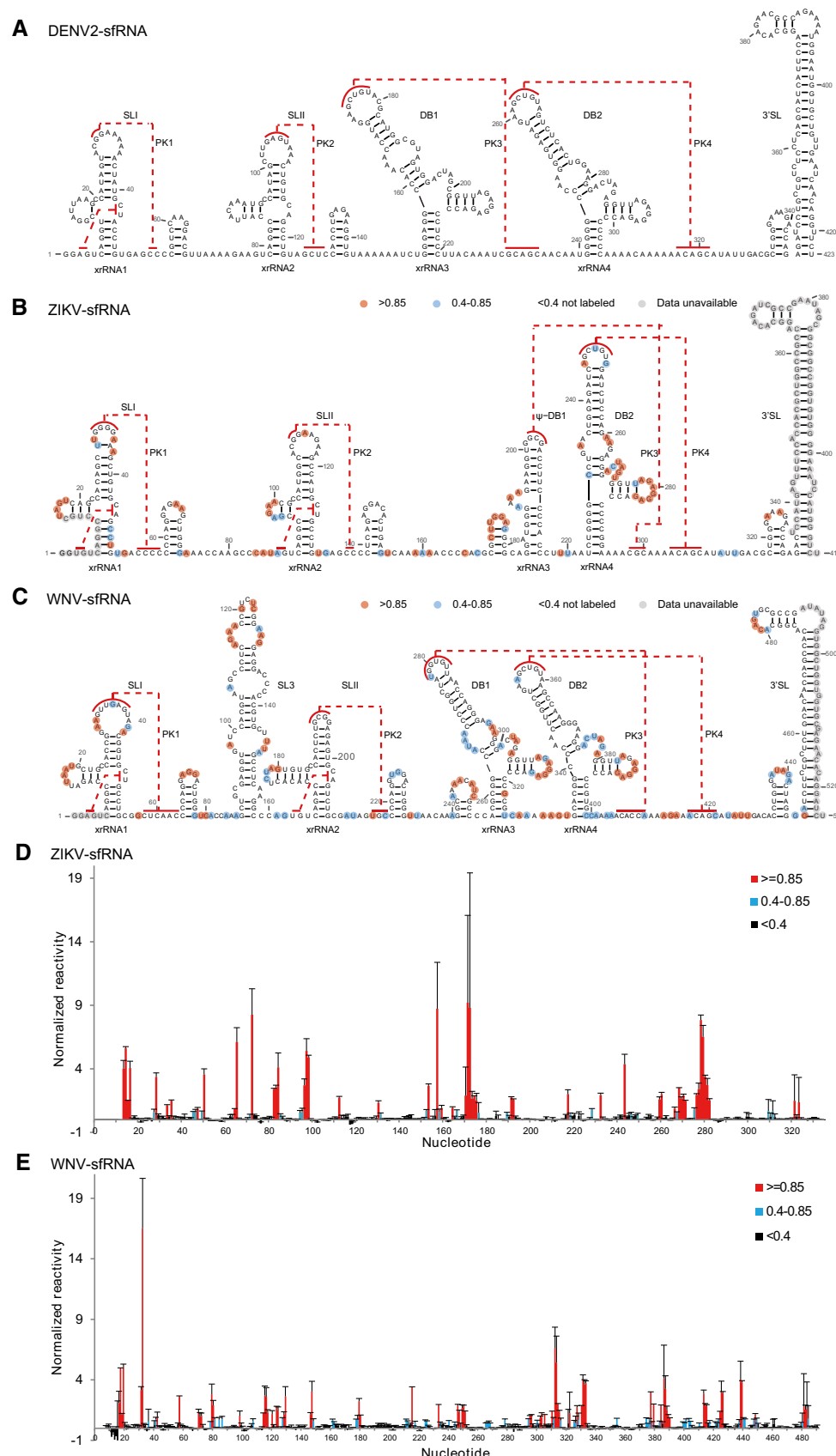

**Figure 1.**

◀

**Figure 1.  Secondary structures of the complete sfRNAs.**

A–C   Secondary structure of the complete sfRNAs of DENV2 (A), ZIKV (B), and WNV (C). The stem-loop structures (SLI and SLII), SL3 of WNV, pseudo-dumbbell (ψ-DB1) structure of ZIKV, duplicated dumbbell structures (DB1, DB2), and the essential 3′ SL are indicated. Sequences involved in pseudoknots (PK1, PK2, PK3, PK4) formation are indicated with red lines.

D, E   SHAPE analysis of the complete sfRNAs of ZIKV (D) and WNV (E). Nucleotides with SHAPE reactivity greater than 0.85 are labeled in red, and moderately reactive site (0.4–0.85) is labeled in cyan. Unlabeled sites have low or no SHAPE reactivity (< 0.4). The SHAPE reactivity was also annotated onto the corresponding secondary structures in (B) and (C). Nucleotides which SHAPE reactivity was not determined are labeled in gray in (B) and (C).

the non-coding viral RNAs: sfRNA1, sfRNA2, sfRNA3, and sfRNA4, respectively, in DENV2-infected human and mosquito cells [14]. Accordingly, the conserved individual RNA subdomains were proposed with functional descriptive names as xrRNA1, xrRNA2, xrRNA3, and xrRNA4, respectively [14]. The sfRNA generation and Xrn1 resistance have also been elucidated for other serotypes of DENV [15], ZIKV [16], WNV [17], Japanese encephalitis virus (JEV) [18], and Murray Valley encephalitis virus (MVEV) [19], and are confirmed to be conserved across flaviviruses [12]. These lncRNAs have been proposed to interact with different viral and host proteins including proteins involved in innate immunity (TRIM25) [20], translation (eEF1A, PABP, La, G3BP1, G3BP2, CAPRIN1) [21], and mRNA metabolism (DDX6) [21]. Although the functions of these lncRNAs have not been fully elucidated, they are implicated to impact on flavivirus replication, cytopathicity, and pathogenicity by modulating multiple cellular pathways, including counteracting type I interferon (IFN) effects, dysregulating RNA decay machinery, and sequestering cellular proteins important in antiviral responses [12,21–23]. Therefore, sfRNA generation has emerged as an interesting target for the development of anti-flavivirus therapeutics or for the rational design of attenuated flavivirus vaccines [21,24].

Despite the availability of abundant functional data, structural studies of these flavivirus lncRNAs, both their subdomains and as a whole, have been limited. Until recently, the crystal structures of xrRNA1 from ZIKV and xrRNA2 from MVEV are determined which reveal that these xrRNAs form multi-pseudoknot structures that resist Xrn1 activity and underlie sfRNA1 and 2 generation [16,25], respectively. Little is known yet about the structure of DB1, DB2, which are also involved in sfRNA generation [14], and of 3′ SL, which is important in viral replication or translation [26]. While the secondary structures of the 3′ UTR of several mosquito-borne flaviviruses (MBFVs) including WNV and ZIKV have been predicted by bioinformatics analysis [14,27], only the secondary structure of the complete sfRNA from DENV2 is interrogated experimentally by selective 2′-hydroxyl acylation analyzed by primer extension (SHAPE) RNA structure probing (Fig 1A) [13]. Furthermore, how the individual subdomains in the context of complete sfRNAs are organized into 3D architecture and their accessible 3D conformational space remains unclear.

A variety of experimental techniques such as SHAPE chemical probing in combination with bioinformatics prediction methods have been developed to characterize the secondary structures of several lncRNAs *in vitro* (such as HOTAIR [28] and SRA [29]) or *in vivo* (such as PAN [4]) [30]; unfortunately, three-dimensional structural characterization of lncRNAs like sfRNAs is still challenging to traditional techniques like X-ray crystallography (XRC), nuclear magnetic resonance (NMR), and cryo-electron microscopy (cryo-EM) [31–33]. Recent progress in small angle X-ray scattering (SAXS) has made it a powerful tool in bridging the gap between the secondary and tertiary structures and characterizing the accessible

3D conformational space of large RNAs in solution [34,35], and SAXS might be the only reasonable method for directly acquiring structural data for large RNAs [36].

Here, we firstly confirmed the previous predicted secondary structures of the flavivirus sfRNAs from ZIKV and WNV with SHAPE probing technique which were then compared with the secondary structure of DENV2 sfRNA [13], revealing significant similarities and differences. In combination with computational modeling and ensemble optimization methods, we built up and refined sets of 3D RNA structures for individual or combined subdomains and characterized the structural ensembles of complete sfRNAs from DENV2, ZIKV, and WNV against SAXS data, thereby providing structural insights into their functions and information on accessible 3D conformational space of sfRNAs in solution. The individual xrRNA1s and xrRNA2s adopt similar tertiary structures in solution as the crystal structure of ZIKV xrRNA1, the xrRNA1, and xrRNA2 in tandem form compact structures with reduced flexibility, suggesting structural basis of functional coupling. While DB12 of DENV2 is extended in solution, the DB12s of ZIKV and WNV are compact due to the formation of intertwined double pseudoknots in proximity, mutations on which affect the RNA structures *in vitro* and hinder viral replication in cell culture. All 3′SLs share similar rod-like structures, indicating coaxial stacking between the small hairpin and the long stem-loop of 3′SL. All the individual subdomains together are organized into elongated, extended conformations in complete sfRNAs that sample large conformational spaces in solution, which may facilitate their binding to different proteins in response to a variety of biological processes. Additionally, our study also highlights the combination of SAXS and computational modeling as ideal tools for exploring the accessible 3D conformational spaces of other lncRNAs in solution.

## Results

### Homogeneity and compactness of the complete sfRNAs

Complete sfRNAs from DENV2, ZIKV, and WNV were prepared by following the native purification protocols which have been successfully used to obtain other lncRNAs in large amount (> 10 mg) [37]. The different sfRNAs migrate as single tight bands on native PAGE gel (Fig EV1A) and behave as monodispersed samples in dynamic light scattering experiments (Fig EV1B), indicating high purity and homogeneity of the samples, which was furthered confirmed by SAXS analysis below.

To identify the optimal ionic conditions that promote the homogeneous compaction of sfRNAs, we studied sfRNA compaction as a function of $Mg^{2+}$ concentration by SAXS. The scattering profiles, with scattering intensity $I(q)$ plotted against momentum transfer $q$, along with pair distance distribution function PDDF transformed

from scattering profiles for DENV2, ZIKV, and WNV sfRNAs at various $Mg^{2+}$ concentrations are shown in Fig EV1C–E. The Guinier regions of all the scattering profiles are linear, indicating that the sfRNAs are monodisperse and homogeneous in solution. The overall structural parameters, including the radius of gyration $R_g$ calculated from Guinier slops, $R_g$, and the maximum diameter $D_{max}$ from PDDF functions, as well as molecular weights derived from volume of correlation ($V_c$) [38], are summarized in Appendix Table S1, among which the molecular weights calculated from SAXS data are consistent with those predicted from sequences, indicating that the complete sfRNAs are all monomeric in our solution conditions. The radius of gyration $R_g$ and the maximum diameter $D_{max}$, plotted against the concentration of $[Mg^{2+}]$ (Fig EV1F and G), reach a minimum at 5 mM, indicating that $Mg^{2+}$ may promote sfRNA compactness at lower concentration but induce attractive intermolecular interactions leading to aggregation at higher concentration. A concentration of 5 mM $Mg^{2+}$ is sufficient to promote proper folding of sfRNAs; we therefore choose this condition for subsequent *in vitro* structural characterization.

**Secondary structures of the complete sfRNAs**

Recently, secondary structures for complete ZIKV and WNV sfRNAs were also proposed based on bioinformatics methods but not experimentally validated (Fig 1B and C), revealing similarities and substantial differences among the subdomains across viruses [14,27]. As shown in Fig 1B and C, both ZIKV and WNV sfRNAs indicate the presence of duplicated SL structures (SLI (xrRNA1) and SLII (xrRNA2)), and the difference is that the xrRNA1 and xrRNA2 in WNV sfRNA are interspersed with an additional long stem-loop structure (SL3). While DENV2 sfRNA contains a duplicated DB structures (DB1 and DB2), ZIKV sfRNA contains only a single copy of DB structure (DB2) which follows a peculiar pseudo-dumbbell structure (ψ-DB1) (Fig 1A and B). WNV sfRNA also possesses duplicated DB structures (DB1 and DB2) (Fig 1C). Interestingly, the topological organizations of the double pseudoknots are different, the single-stranded sequences involved in PK3 and PK4 formation in both ZIKV and WNV sfRNAs locate downstream of DB2 structure (Fig 1B and C), resulting in intertwined double pseudoknots, and in contrast, the sequences involved in PK3 and PK4 formation in DENV2 sfRNA are located before and after DB2 structure, respectively (Fig 1A). All sfRNAs contain the highly conserved 3′ SL structures consisting of a small hairpin (sHP) and a long stem-loop (SL) structure.

We confirmed the predicted secondary structures of ZIKV and WNV sfRNAs by SHAPE chemical probing, which the SHAPE reagent N-methylisatoic anhydride (NMIA) selectively acylates the 2′-hydroxyl group of nucleotides in single-stranded or flexible regions of RNA [39]. The SHAPE reactivity is monitored at single nucleotide resolution (Fig 1D and E). The normalized reactivity

data are annotated onto the secondary structure models (Fig 1B and C). All the nucleotides exhibiting high reactivities are located at loops or single-stranded regions, and all nucleotides involved in base-pairing or potential pseudoknot formation have low reactivities, indicating the high consistency of our SHAPE data with the secondary structure models. It is interesting to note that the single-stranded linker region between xrRNA1 and xrRNA2 of ZIKV sfRNA shows reduced SHAPE reactivities (Fig 1B), suggesting a high degree of structural restraints which will be discussed later.

***Ab initio* modeling fails to define unique 3D topological structures for the complete sfRNAs**

To determine how the individual subdomains of sfRNAs are organized into the three-dimensional structures, we firstly conducted SAXS experiments on the complete sfRNAs. The experimental SAXS curves, the PDDFs, the dimensionless Kratky [40], and Porod-Debye plots [41] for the complete sfRNAs in the presence of 5 mM $Mg^{2+}$, are shown in Fig 2A–D. The scattering curves in the high-$q$ region have fine features typical of nucleic acids, such as the P1 peak around $q$ of 0.5 $Å^{-1}$, albeit with attenuated peak intensity compared to those of a simple duplex [42] (Fig 2A). This is likely due to the presence of non-duplex structure elements as well as dynamical conformational averaging within sfRNAs. The PDDFs for DENV2, ZIKV, and WNV sfRNAs are quite different at first glance but all show two main distance distributions, one common distance at ~25 Å, characteristic distance within an A-form RNA duplex [42], and the others at around 80, 76, 48 Å for sfRNAs from DENV2, ZIKV, and WNV, respectively, which are all shorter than half the maximum distances ($D_{max}$) within the molecules, indicating that all sfRNAs are rather elongated in solution (Fig 2B). The dimensionless Kratky plots, plotted as $(qR_g)^2 I(q)/I(0)$ vs. $qR_g$, showing double peaks and larger peak positions in comparison with a single peak at lower $qR_g$ for ZIKV xrRNA1, which is a compact, well-folded RNA, suggest that sfRNAs are grafted, extended molecules in solution (Fig 2C). The Porod-Debye plots, plotted as $q^4I(q)$ vs. $q^4$, lack obvious plateau at low $q$ region in comparison with ZIKV xrRNA1, indicating increased flexibility in the sfRNAs (Fig 2D). These structural features are direct observables and not subject to possible bias due to limitation of software and therefore are important in guiding the interpretation presented and discussed below.

To gain more specific information on the overall 3D structures of the complete sfRNAs, we initially attempted to reconstruct their *ab initio* shape envelops using the program DAMMIN and auxiliary programs [43]. Using SAXS data with a $q_{max}$ of about 0.3 $Å^{-1}$, this strategy has been employed to determine the shape structures of several large structured RNAs [44,45] at the resolution of A-form helices (~20 Å) which models a macromolecule as an assembly of scattering beads arranged in space such that a calculated scattering

**Figure 2. Overall structural analysis of the complete sfRNAs by SAXS.**

A–D    The scattering profiles (A), pair distance distribution functions (PDDFs) (B), dimensionless Kratky plots (C), and Porod-Debye plots (D) of the complete sfRNAs of DENV2 (cyan line), ZIKV (red line), and WNV (blue line). The fine features of P1 in the scattering profiles in (A) arise from helical interstrand pair distance correlation. The PDDF profiles in (B) were calculated using GNOM ($q_{max}$ = 0.3). The dimensionless Kratky plot and Porod-Debye plot for ZIKV xrRNA1 (red dashed line) were included in (C) and (D) for comparison, indicating that the structures of sfRNAs are more extended and open than that of ZIKV xrRNA1.

E       The *ab initio* shape envelopes of the complete sfRNAs shown in three views. The spatial resolution of the envelopes is 21 Å.

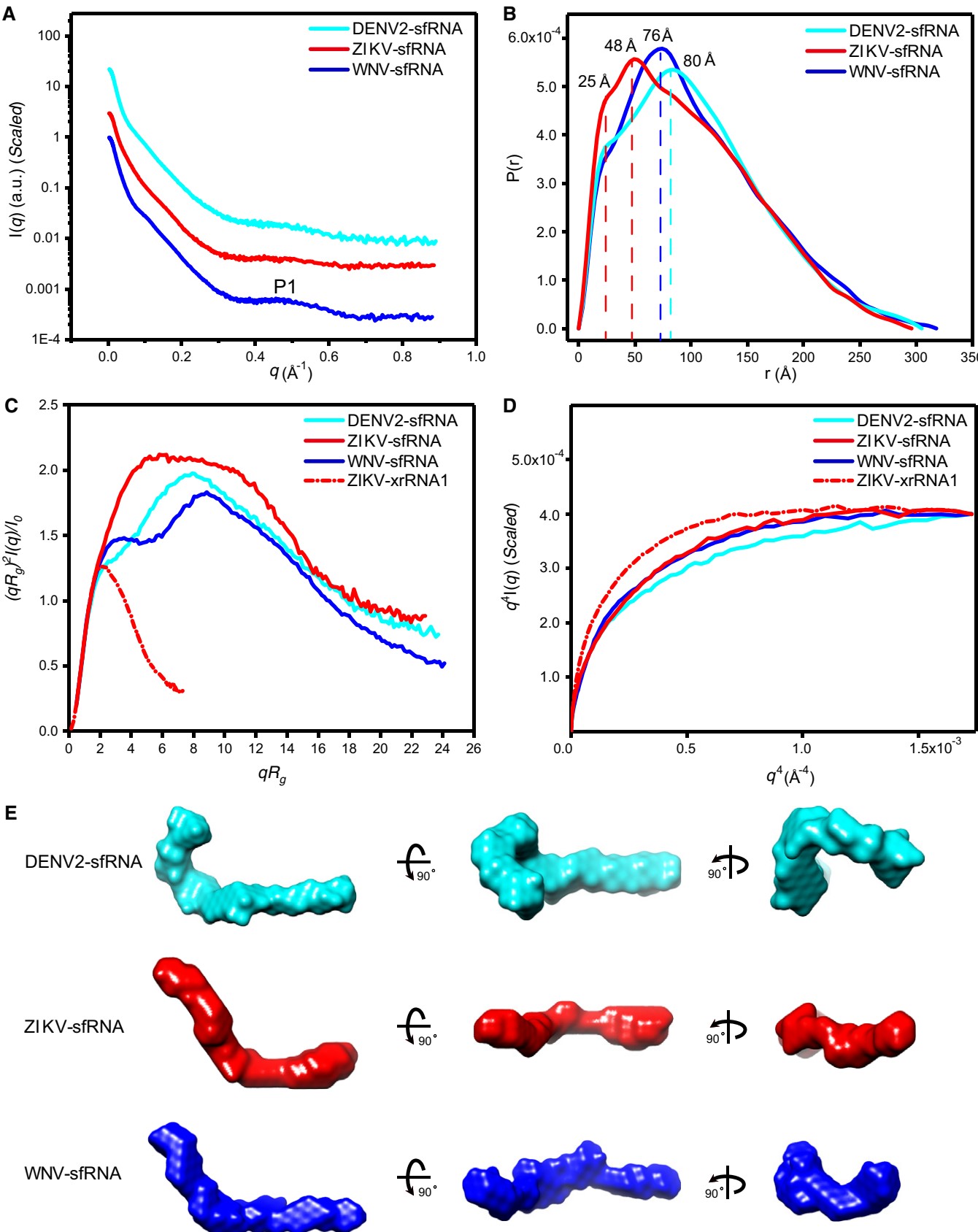

**Figure 2.**

curve reproduces the experimental curve. The resulting averaged shape envelopes shown in Fig 2E are all elongated, indicating that all sfRNAs are extended and open molecules in solution. However, the lack of features of these envelopes that would be instructive for structural interpretation at the resolution of A-form helices prevents direct localization of the individual subdomains. One reason for such an outcome could be the increased intrinsic conformational flexibility due to their large sizes, which are also evidenced by the normalized spatial discrepancy (NSD) scores of 1.141, 0.975, and 1.069 during the shape reconstruction for complete sfRNAs of DENV2, ZIKV, and WNV, respectively. NSD is a quantitative measure of the differences in spatial occupancy between each shape [46]. For ideally superimposed multiple reconstructions, NSD tends to zero whereas it exceeds unity when the individual aligned solutions systematically differ. Therefore, despite excellent quality of sample homogeneity and SAXS data, NSD values tending to one (or more) can be associated with conformational heterogeneity and so suggest that all sfRNA sample diverse conformations in solution (Fig EV2). As all conformations in solution contribute to the overall scattering of the molecules, such conformational heterogeneity would blur the feature of the scattering curves (Fig 2A) as well as the appearance of the averaged shape envelopes (Fig 2E), therefore preventing from defining unique 3D topological structures for the complete sfRNAs directly.

As an alternative to *ab initio* shape reconstruction for topological structure determination of a molecule, 3D all-atom models of large RNAs can be constructed by incorporating the known atomic models of individual subdomains and iterative refining against the experimental SAXS data [47]. We therefore dissect the sfRNAs into its individual subdomains or combination first which are further analyzed by SAXS.

### Solution structures of the individual xrRNA1s and xrRNA2s

As shown in Fig 1A–C, the individual xrRNA1s and xrRNA2s share similar secondary structures which consist of a three-way junction formed by P1, P2, and P3, and an additional P4 helix. Two crystal structures were recently determined for the xrRNA1 from ZIKV and xrRNA2 from MVEV, respectively, which reveal high similarity in the core regions formed by the three-way junctions (Fig 3A), but significant differences in the P4-L4 stem-loop (Fig 3B). The formation of L3-S4 pseudoknot in ZIKV xrRNA1 places the P4-L4 stem-loop in a different position relative to the core region from that in the MVEV xrRNA2 structure [25], which the L3-S4 pseudoknot is absent (Fig 3B).

To analyze their 3D structures in solution, we carried out SAXS experiments on xrRNA1s and xrRNA2s from DENV2, ZIKV, WNV,

and MVEV, respectively. Inspection of the scattering profiles and linearity of the Guinier regions confirms monodispersity and absence of aggregation in all individual xrRNA1s and xrRNA2 in solution (Fig 3C and D). The asymmetric PDDF functions indicate a slightly elongated molecule with asymmetric shape (Fig 3C and D). The overall structural parameters are summarized in Appendix Table S2. The molecular weight calculated from SAXS data consistently points to a monomeric molecule for each construct in solution (Appendix Table S2). The dimensionless Kratky plots and Porod-Debye plots suggest that all individual xrRNA1s and xrRNA2s are well-folded and of reduced flexibility in solution (Fig 3C and D). The *ab initio* shape reconstructions for xrRNA1s and xrRNA2s reveal similar asymmetric global envelopes in solution (Fig 3E and F). The high quality and reproducibility of the resulting envelopes are confirmed based on the calculated chi-square of fitting and NSD values, respectively (Appendix Table S2). The back-calculated scattering profile of ZIKV xrRNA1 fits well to its experimental curve ($\chi^2 = 0.82$) (Fig 3C), indicating a very similar conformation between SAXS condition and in crystal, in contrast, the back-calculated scattering profile for MVEV xrRNA2 fits poorly to its experimental curve ($\chi^2 = 14.66$) (Fig 3D), and therefore, MVEV xrRNA2 in solution must adopt a conformation significantly different from that observed in crystal lattice.

The primary sequences and potential secondary structures of xrRNA1 and xrRNA2 across mosquito-borne flavivirus (MBFV) are highly conserved (Fig EV3A). To obtain more detailed 3D conformational images, homology models are built for xrRNA1s and xrRNA2s using ModeRNA [48], which is a program of comparative modeling of RNA 3D structures that requires a pairwise sequence alignment and a structural model as inputs. The crystal structure of ZIKV xrRNA1 is used as the template for modeling. All the generated models contain ring-like core regions and the formation of L3-S4 pseudoknots which is further stacked by the P4-L4 stem-loop (Fig 3E and F). All the back-calculated scattering curves of the generated models can nicely fit to the experimental scattering curves (Fig EV3B), suggesting high confidence of the homology models representing solution structures of individual xrRNA1s and xrRNA2s and the formation of L3-S4 pseudoknots in MVEV xrRNA2 which is absent in its crystal structure. The major differences lie in the L2-P2 stem-loops, which vary in length and are the least conserved in the secondary structures (Fig 3E and F). The resulting models can also be nicely fitted into the corresponding *ab initio* shape envelopes (Fig 3E and F). The convergence between the global shape of the envelope and the all-atom models therefore provides further confidence in the overall conformation of xrRNA1s and xrRNA2s in solution.

---

**Figure 3. SAXS analysis of the individual xrRNA1s and xrRNA2s.**

A	Crystal structures of ZIKV xrRNA1 (left, PDB code: 5TPY) and MVEV xrRNA2 (right, PDB code: 4PQV).

B	Two views of the overlay of ZIKV xrRNA1 and MVEV xrRNA2 crystal structures.

C, D	The scattering profiles (left), the PDDFs (middle), and the dimensionless Kratky plots (right) of individual xrRNA1s (C) and xrRNA2s (D) of DENV2 (cyan), ZIKV (red), WNV (blue), and MVEV (green). The insets in (C) and (D) show the Guinier regions of the respective scattering profiles with a linear fit line. The back-calculated scattering profile of ZIKV-xrRNA1 crystal structure (black, open circle) can be nicely fitted onto its experimental SAXS data (red line) in (C), while the back-calculated scattering profile of MVEV-xrRNA2 crystal structure (black, open circle) fits poorly to its experimental SAXS data (green line) in (D). The asymmetric PDDFs of xrRNA1 (C) and xrRNA2 (D) indicate slightly elongated molecules with asymmetric shapes. The dimensionless Kratky plots of xrRNA1 (C) and xrRNA2 (D) of DENV2, ZIKV, WNV, and MVEV suggest all individual xrRNA1s and xrRNA2s are well-folded and of reduced flexibility in solution.

E, F	The *ab initio* shape envelopes of the individual xrRNA1s (E) and xrRNA2s (F) of DENV2 (cyan), ZIKV (red), WNV (blue), and MVEV (green) are fitted with the homologous atomic models built by ModeRNA.

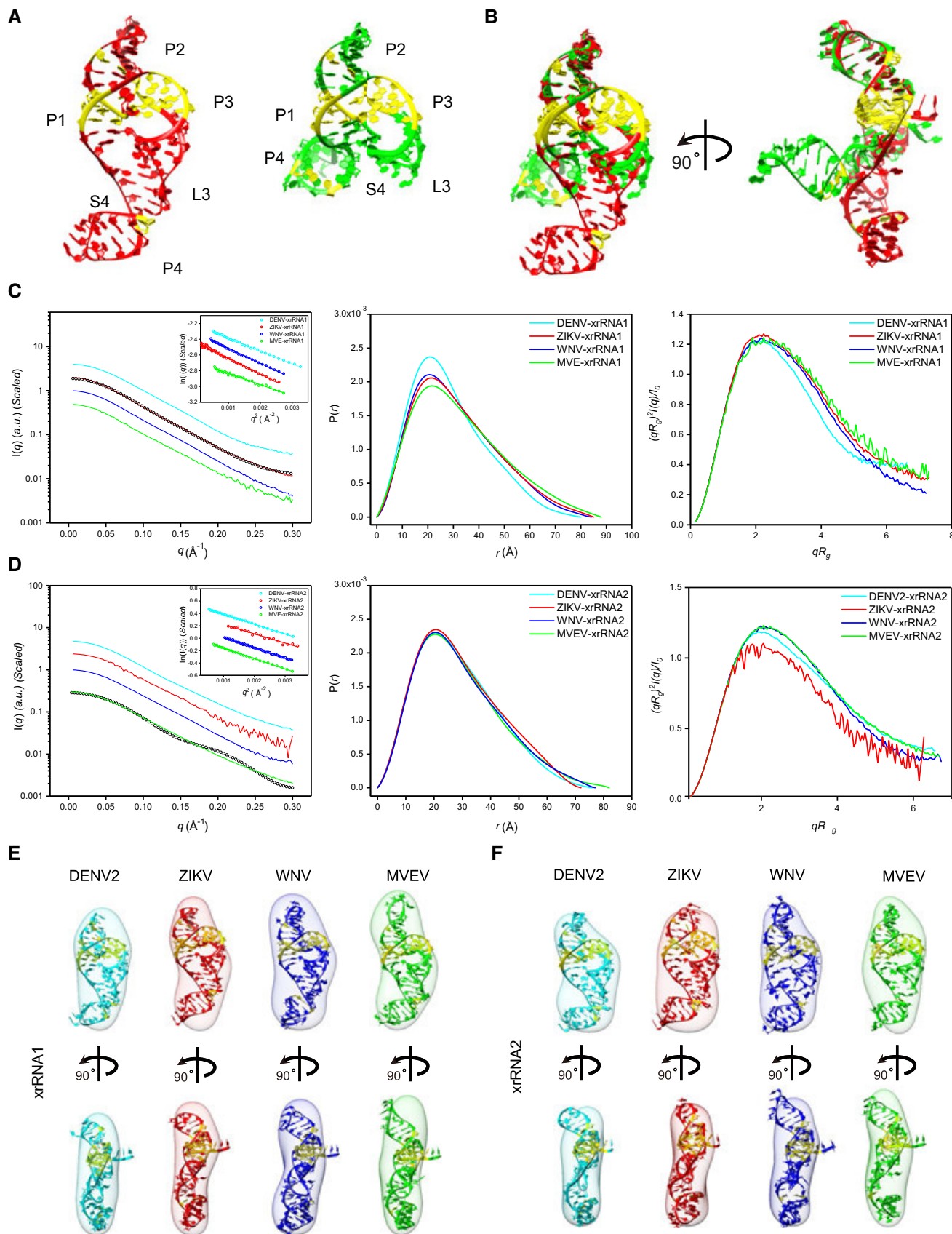

**Figure 3.**

## Solution structures of xrRNA1 and xrRNA2 in tandem

An intriguing feature in most mosquito-borne flavivirus (MBFV) RNA genomes is the presence of duplicated SL structures (xrRNA1 and xrRNA2) in the 3′ UTR [49]. Several experimental evidences support close functional couplings between xrRNA1 and xrRNA2 in both WNV and DENV2 recently [13,17,25,50] that sfRNA1 production by xrRNA1 is coupled to the integrity of the tertiary structure of xrRNA2 and vice versa, suggesting potential structural coupling between xrRNA1 and xrRNA2. We therefore analyzed the solution structures of xrRNA1 and xrRNA2 in tandem (hereafter abbreviated as xrRNA1-2) from DENV2, ZIKV, and WNV using SAXS. It is necessary to point out that xrRNA1 and xrRNA2 of WNV are intercalated with a stem-loop structure SL3.

The respective scattering profiles, PDDF functions of xrRNA1-2s, are shown in Fig 4A and B, and the overall structural parameters are summarized in Appendix Table S2. Inspection of the scattering profiles and the Guinier fittings indicate that all the xrRNA1-2s are monodisperse and free of aggregation in solution (Fig 4A). The PDDFs of xrRNA1-2 of DENV2 and ZIKV are similar in general but quite different from that of WNV xrRNA1-2, suggesting quite different structure of WNV xrRNA1-2 (Fig 4B). The molecular weight calculated from SAXS data is consistent with a monomeric state for each construct in solution (Appendix Table S2). All the dimensionless Kratky plots are characteristic of well-folded molecules, suggesting reduced flexibility of xrRNA1-2 (Fig 4C). The double peaks in the Kratky plot of WNV xrRNA1-2 are characteristic of branched RNA, which is confirmed by *ab initio* shape reconstructions using DAMMIN. As shown in Fig 4D, the globe shape envelope for WNV

xrRNA1-2 branches in the middle, different from the elongated overall shape envelopes for DENV2 and ZIKV xrRNA1-2. The NSD scores of the DAMMIN models for DENV2-, ZIKV- and WNV-xrRNA1-2s are 0.83, 0.78, and 0.77, respectively, suggesting high quality and reproducibility of the respective envelops (Appendix Table S2).

As no high-resolution atomic model is available for SL3 of WNV, an ensemble of all-atom models is firstly predicted for SL3 with the *de novo* RNA structure predication program Rosetta, which follows a two-step protocol named as FARFAR (Fragment assemble of RNA with Full Atom Refinement) [51]. The resulting models are screened against SAXS data, resulting in a best fit model with $\chi^2$ of 0.87 that represent SL3 in solution (Fig EV4).

With the availability of atomic models for all xrRNA1s, xrRNA2s, and SL3, rigid-body modelings are further performed to build up atomic models for the respective xrRNA1-2s using the program Xplor-NIH [52,53], during which the individual xrRNA1s, xrRNA2s, and SL3 are treated as rigid bodies, the linkers linking xrRNA1, xrRNA2, and SL3 are allowed to move freely, and the relative positions and orientations of the individual domains are refined against the SAXS data by a simulated annealing algorithm [54]. The best fit all-atom models, with best fitting $chi^2$ of 0.44, 0.20, and 0.15 for DENV2-, ZIKV-, and WNV-xrRNA1-2, respectively, can be nicely superimposed onto the respective *ab initio* shape envelopes, as shown in Fig 4D. While the best fit models for DENV2 and ZIKV xrRNA1-2s suggest compact conformations and relative orientations between xrRNA1 and xrRNA2, in support of strong structural coupling, the SAXS-derived best fit model for ZIKV xrRNA1-2 shows a topological structure significantly different from what has been

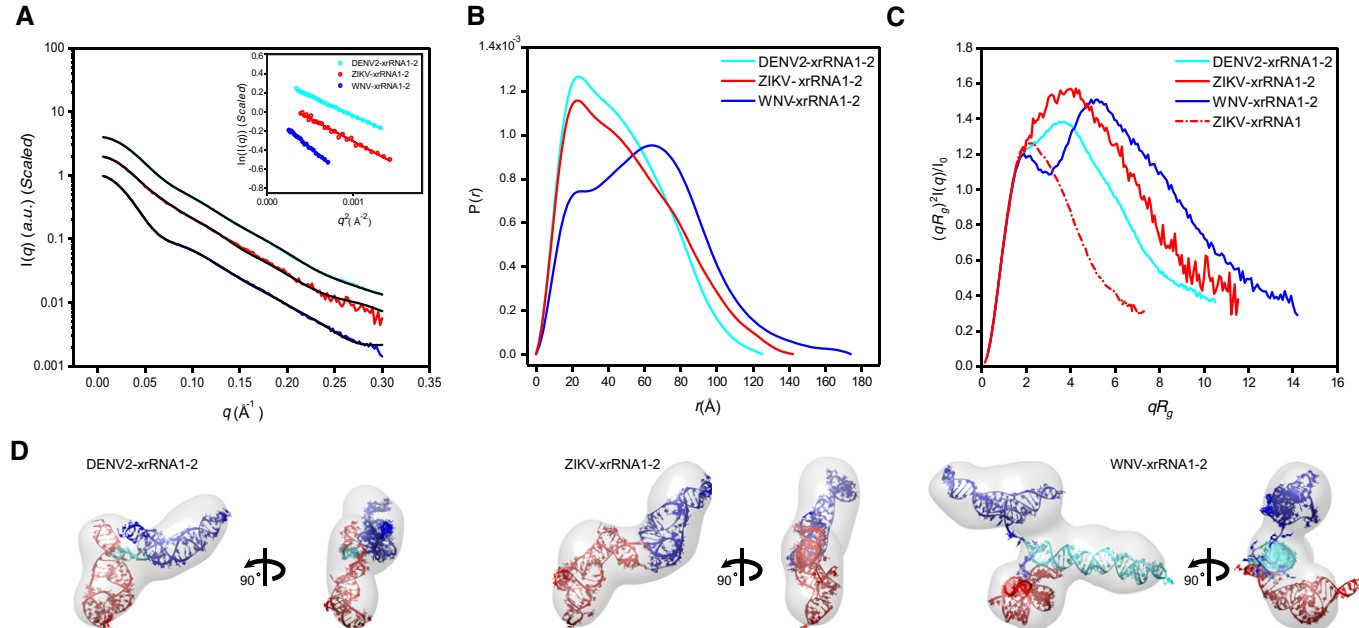

**Figure 4. SAXS analysis of the xrRNA1-2s.**

A–C The scattering profiles (A), the PDDFs (B), and the dimensionless Kratky plots (C) of xrRNA1-2s from DENV2 (cyan), ZIKV (red), and WNV (blue). The inset in (A) shows the guinier regions of the respective scattering profiles with linear fit lines. The dimensionless Kratky plot of ZIKV xrRNA1 (red dashed line) was included in (C) for comparison, indicating that the structures of xrRNA1-2s are all well-folded.

D The *ab initio* shape envelopes of the xrRNA1-2s of DENV2 (left), ZIKV (middle), and WNV (right). The atomic models from rigid-body modeling by Xplor-NIH are superimposed onto the respective envelopes. The back-calculated scattering profiles of the atomic models were fitted to the respective experimental scattering profiles in (A).

predicted solely based on sequence earlier [55]. In the best fit model of WNV xrRNA1-2, SL3 sticks out of the relative compact folding of xrRNA1 and xrRNA2 in tandem, consistent with what has been predicted earlier [27]. Within the resolution restrictions of SAXS experiments, it is interesting to note that the SAXS-derived models for xrRNA1-2s from DENV2 and ZIKV share similar organization between xrRNA1 and xrRNA2.

### Solution structures of DB1 and DB2 in tandem

The presence of duplicated dumbbell (DB) structures is another conserved common feature in most of mosquito-borne flaviviruses 3′ UTRs [49]; currently, there is no high-resolution tertiary structure information available for any of the DB1 and DB2 structures. The

organization of the DB secondary structures in DENV2, ZIKV, and WNV 3′ UTR varies greatly (Fig 1A–C), whether and how these organization patterns affect the 3D structures of DB1 and DB2 remains unclear. To this end, we characterized the solution structures of DB1 and DB2 in tandem (hereafter abbreviated as DB12) from DENV2, ZIKV, and WNV, respectively, using SAXS.

The respective scattering profiles and PDDF functions of DB12s from DENV2, ZIKV, and WNV are shown in Fig 5A and B, and the overall structural parameters are summarized in Appendix Table S2. Inspection of the scattering profiles and Guinier regions indicate that all the DB12s are monodisperse and free of aggregation in solution. The molecular weights derived from SAXS data suggest all the molecules are monomeric in solution. While the dimensionless Kratky plots for the DB12s from WNV and ZIKV are similar in general and

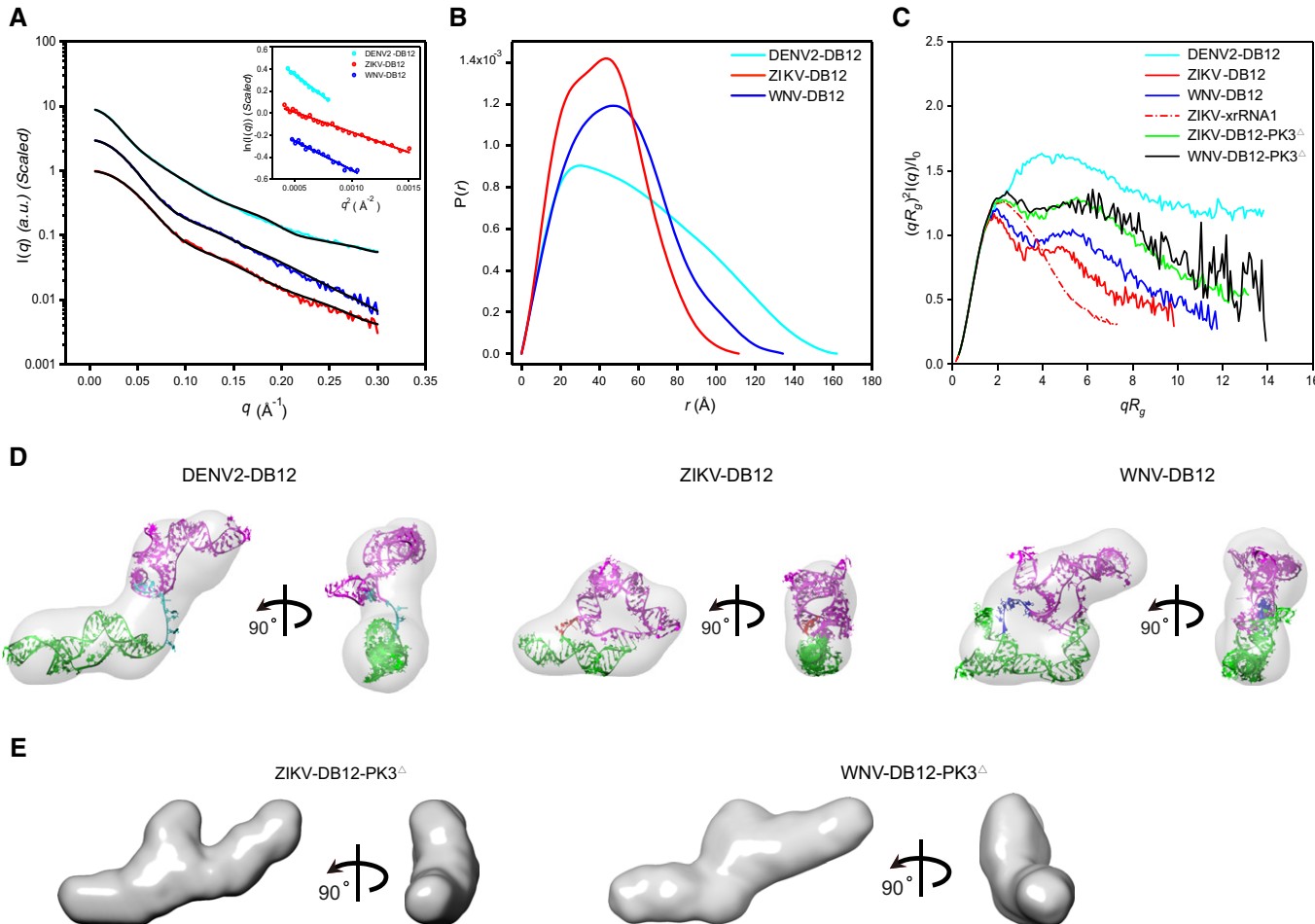

**Figure 5. SAXS analysis of the DB12s.**

A–C  The scattering profiles (A), the PDDFs (B), and the dimensionless Kratky plots (C) of DB12s from DENV2 (cyan), ZIKV (red), and WNV (blue). The inset in (A) shows the Guinier regions of the respective scattering profiles with linear fit lines. The dimensionless Kratky plot of ZIKV xrRNA1 (red dashed line) was included in (C) for comparison, indicating that the structures of DB12s of ZIKV and WNV are well-folded, but that of DB12 of DENV2 is partially folded. The Kratky plots for the two PK3 mutants of DB12s, the ZIKV-DB12-PK3$^\Delta$ (green) and WNV DB12-PK3$^\Delta$ (black), respectively, are also included for comparison, suggesting reduced compactness upon PK3 mutations.

D  The *ab initio* shape envelopes of the DB12s of DENV2 (left), ZIKV (middle), and WNV (right) in two views. The atomic models from *de novo* structure modeling by Rosetta were superimposed onto the respective envelopes. The back-calculated scattering profiles of the *de novo* atomic models are fitted to the respective experimental scattering profiles in (A).

E  The *ab initio* shape envelopes of the PK3 mutants of ZIKV DB12s, the ZIKV-DB12-PK3$^\Delta$ (left) and WNV DB12-PK3$^\Delta$ (right) in two views, suggesting open conformations upon PK3 mutations.

characteristic of well-folded compact molecules, the dimensionless Kratky plot suggests that DB12 from DENV2 is partially folded or a multi-domain RNA with flexible linkers (Fig 5C). The Kratky analysis is consistent with *ab initio* shape reconstruction which generates an elongated and open envelope for the DB12 from DENV2, but the shape envelopes for DB12s of ZIKV and WNV are compact and closed (Fig 5D). The low chi-square value of fitting to experimental curves and small NSD values indicate high quality and good reproducibility of the *ab initio* shape envelopes (Appendix Table S2).

As no homologous high-resolution structures are available yet, based on the available secondary structure information, ensembles of *de novo* all-atom models were generated for all the DB12s using Rosetta [51] and screened to fit respective SAXS data, with the best fitting $\chi^2$ values of 1.52, 0.15, and 0.14 for DENV2, ZIKV, and WNV, respectively. The best fit models can be nicely fitted into the respective *ab initio* envelopes of the DB12s (Fig 5D). In each of the selected models, both PK3 and PK4 are formed. The differences and similarities in the shape envelopes can be explained by the topological organization of PK3 and PK4. In DB12 of DENV2, the PK3 and PK4 formation are in sequential order, thus resulting in an extended conformation. In ZIKV- and WNV DB12s, the PK3 and PK4 are in proximity and intertwined, facilitating closed and compact conformations (Fig 5D). The formation and importance of the PK3 in ZIKV and WNV DB12s are further confirmed and evaluated by mutagenesis analysis. Disruption of PK3 in both ZIKV and WNV DB12s leads to mutants (ZIKV-DB12-PK3$^\Delta$: $^{200}$GC$^{202}$G→$^{200}$CG$^{202}$C, and WNV DB12-PK3$^\Delta$: $^{279}$GGUG$^{283}$U →$^{279}$CCAC$^{283}$A, respectively; Appendix Table S4) with reduced compactness, as evidenced by the Kratky plots in Fig 5C, the open and extended shape envelopes in Fig 5E, and increased $R_g$ (Appendix Table S2), suggesting important roles of the PK3s in promoting the proper folding of the ZIKV and WNV DB12s.

## Biological significance of the DB12 intertwined pseudoknots in ZIKV

The biological significance of the DB12 intertwined pseudoknots in ZIKV was further evaluated by mutational analysis based on replicon and infectious clone system. Three mutants, ZIKV-PK3$^\wedge$, ZIKV-PK3*, and ZIKV-DENV+DBs, were generated based on the infectious cDNA clone of ZIKV strain FSS13025 [56], respectively. In ZIKV-PK3$^\wedge$, the corresponding sequence of $^{298}$CG$^{300}$C in ZIKV sfRNA was replaced with $^{298}$AG$^{300}$A to disrupt the original PK3 structure (Fig 6A). The ZIKV-PK3* was constructed on the basis of ZIKV-PK3$^\wedge$, in which "AACGCUU" was inserted between $^{220}$U and $^{221}$A to form a DENV2-like DBs (Fig 6B). In ZIKV-DENV+DBs, the DB12 sequence from $^{183}$A to $^{308}$C in ZIKV sfRNA (Fig 1B) was replaced with the corresponding DB12 sequence from $^{152}$U to $^{322}$A in DENV sfRNA (Fig 1A). The *in vitro* transcribed ZIKV RNA is transfected into the BHK-21 cells, and we firstly show that all the ZIKV mutants are non-lethal by the indirect immunofluorescence assay (IFA, Fig 6C). However, compared to the wild-type ZIKV, the viral protein expression of ZIKV-PK3$^\wedge$ and ZIKV-PK3* is significantly reduced at 24–72 h post-transfection. Interestingly, the level of viral expression in the BHK-21 cells transfected with the ZIKV-DENV+DBs mutant is almost the same as the wild type. Further, the yields of progeny viral RNAs in the culture supernatants are determined by qRT–PCR, in consistent with the results of immunostaining assay, and the progeny viral RNA copies of ZIKV-PK3$^\wedge$ and ZIKV-PK3* are much lower than

wild-type ZIKV, while the corresponding levels of the ZIKV-DENV+DBs are not changed compared with the wild type (Fig 6D).

Additionally, the same nucleotide substitution and insertion as ZIKV-PK3$^\wedge$ and ZIKV-PK3* are also introduced into the ZIKV replicon (ZIKV-Rep) with an Renilla luciferase reporter, resulting in two mutants of Rep-PK3$^\wedge$ and Rep-PK3*, respectively. A mutant, Rep-GAA, which contains a GDD to GAA mutation in the catalytic motif of the NS5 RdRp domain, is included as the negative control. Routine replicon assay shows that at 36 h post-transfection, the relative luciferase units of the Rep-PK3$^\wedge$ and Rep-PK3* are 5 and 8 times lower than ZIKV-Rep, respectively, while the Rep-GAA exhibits no viral replication as expected (Fig 6E). Together, these data suggest that alternation of the PK3 structure reduced the replication and infectivity of ZIKV, highlighting the biological importance of the intertwined organization of the ZIKV DBs.

## Solution structures of the 3′SLs

The 3′SL structure, consisting of a small hairpin (sHP) followed by a long stem-loop, is highly conserved among all flavivirus genomes [10]. A previous *in vitro* study suggested a potential pseudoknot formation between 4 nt in the loop of sHP and 4nt on the 5′ side of the longer stem of the WNV 3′ SL structure [57], which, however, was not supported by a recent NMR study [58]. We therefore studied the 3′SL structures of DENV2, ZIKV, and WNV sfRNAs in solution using SAXS, which could provide insight into the possibility of potential tertiary interactions.

The scattering profiles, PDDF functions, and dimensionless Kratky plots for the 3′SL structures of DENV2-, ZIKV- and WNV sfRNAs in solution are shown in Fig 7A–C. The overall structural parameters are summarized in Appendix Table S2. The scattering curves and Guinier plots show no evidence of aggregation, and the molecular weights calculated from SAXS data are consistent with monomers in solution. The dimensionless Kratky plots indicate the 3′SLs are all well-folded in solution. The PDDF functions are characteristic of rod-like structures, which are further supported by *ab initio* shape reconstructions which results in elongated rod-like envelopes (Fig 7D), suggesting that the sHP and the longer stem-loop of 3′SL structures are likely coaxially stacked. The high quality and reproducibility of the shape envelopes are confirmed by the best fitting to the experimental scattering curves and reduced NSD values among models (Appendix Table S2).

An ensemble of all-atom models is generated for corresponding 3′SL constructs using the program Rosetta and screened against the SAXS data (Fig 7D). Superimposing the best fit models onto the *ab initio* shape envelopes clearly show that both methods converge on structures in which the two stem-loops of 3′ SL structures are coaxially stacked (Fig 7D), and therefore, the potential tertiary interactions are unlikely formed. This is further supported by SAXS analysis of a mutant of WNV 3′SL (WNV-3′SLM: $^{438}$UAG$^{441}$A→$^{438}$GUC$^{441}$U), as shown in Fig 7A–D; the scattering curve, the PDDF, the Kratky plot, and the shape envelope of WNV 3′SL are only slightly affected by the mutation.

## Structural ensembles of the complete sfRNAs in solution

With the availability of the above 3D atomic models for the respective individual subdomains or combinations, 3D atomic models are generated for the complete sfRNAs with the rigid-body modeling

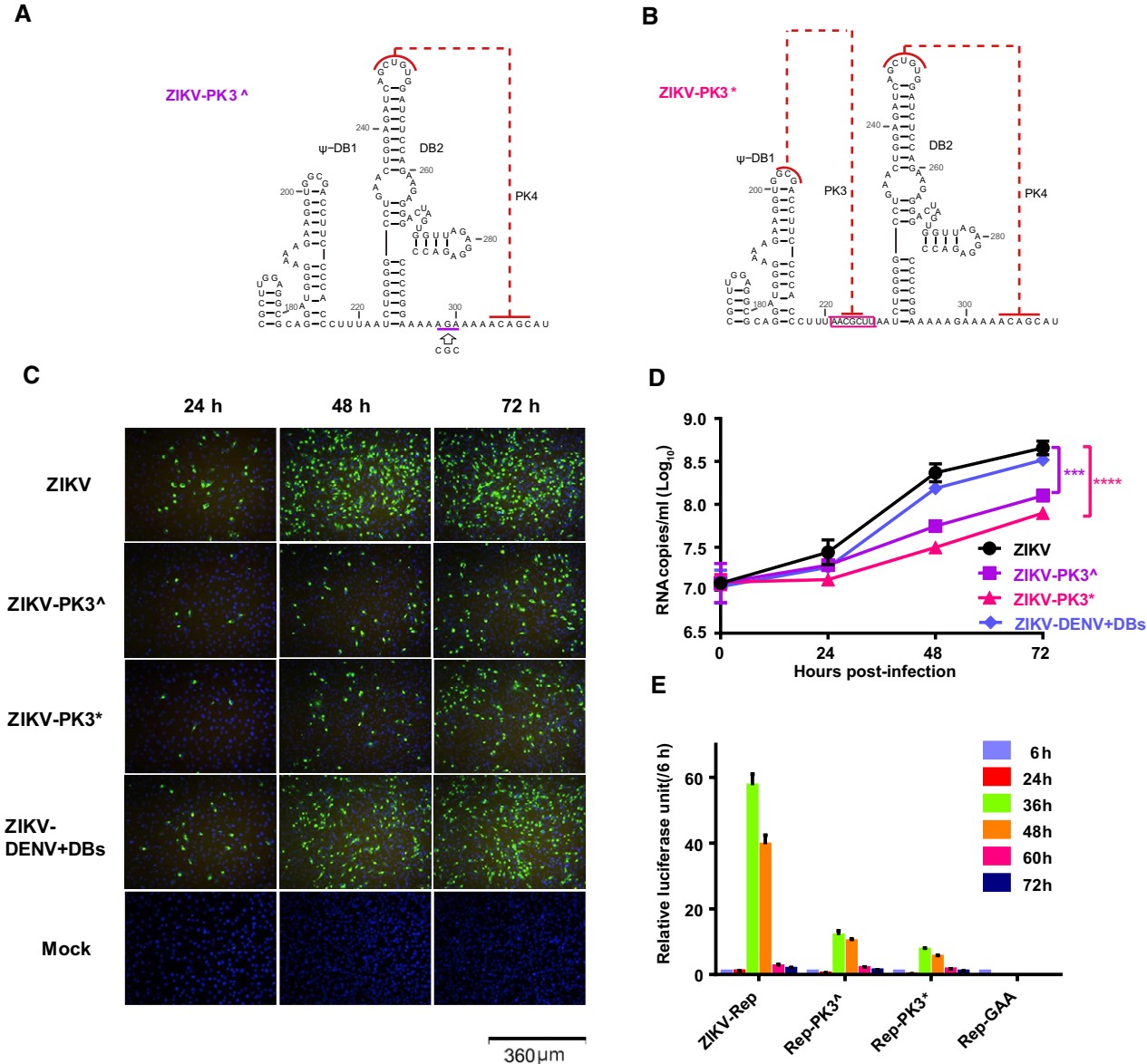

**Figure 6. Mutational analysis on the relevance of DB12 pseudoknots in ZIKV replication.**

A, B Mutational schemes of the two PK3 mutants, ZIKV-PK3^ (A) and ZIKV-PK3* (B), mapping to the secondary structures of the DB12 region.
C E protein expression of ZIKV and the mutants showed by IFA.
D The progeny virus RNA in cultured supernatants of ZIKV and the mutants was detected by qRT–PCR in three technical replicates. Two-way ANOVA and Dunnett's multiple comparison test were used for statistical analysis. The error bars represent the standard deviation. ***$P < 0.001$, ****$P < 0.0001$.
E The luciferase activity of the ZIKV-Rep and the mutants was measured at different time points after transfection in three technical replicates. The error bars represent the standard deviation.

algorithm (Fig EV5A–C) and fitted against the experimental SAXS scattering data using Xplor-NIH (Fig EV5D–F). The best fit models of the complete sfRNAs are all elongated and extended in one dimension consistent with the *ab initio* bead modeling (Fig 2E). The SAXS-derived models for WNV- and ZIKV sfRNAs are significantly different from the models predicted earlier solely based on sequences which are relatively compact [27,55]. The conformations of sfRNAs may be mediated by the flexible single-stranded sequences between subdomains. For example, in Fig 1B, the sequence between SLII and ψ-DB1 of ZIKV sfRNA exhibits high

reactivities, indicating increased flexibility. Targeting this region of ZIKV sfRNA by a complementary locked nucleic acids (LNA) leads to overall conformational changes, as evidenced by obvious shift retardation in native PAGE gel, significant differences in scattering profile, PDDF, and overall shape envelopes, larger $R_g$ and $D_{max}$ (Fig EV5G–J, Appendix Table S1).

Due to the intrinsic flexibility, large RNAs naturally exist as dynamic structural ensembles in solution. To analyze the conformational flexibility of the respective complete sfRNAs and the conformational spaces of the respective sfRNA sample in solution,

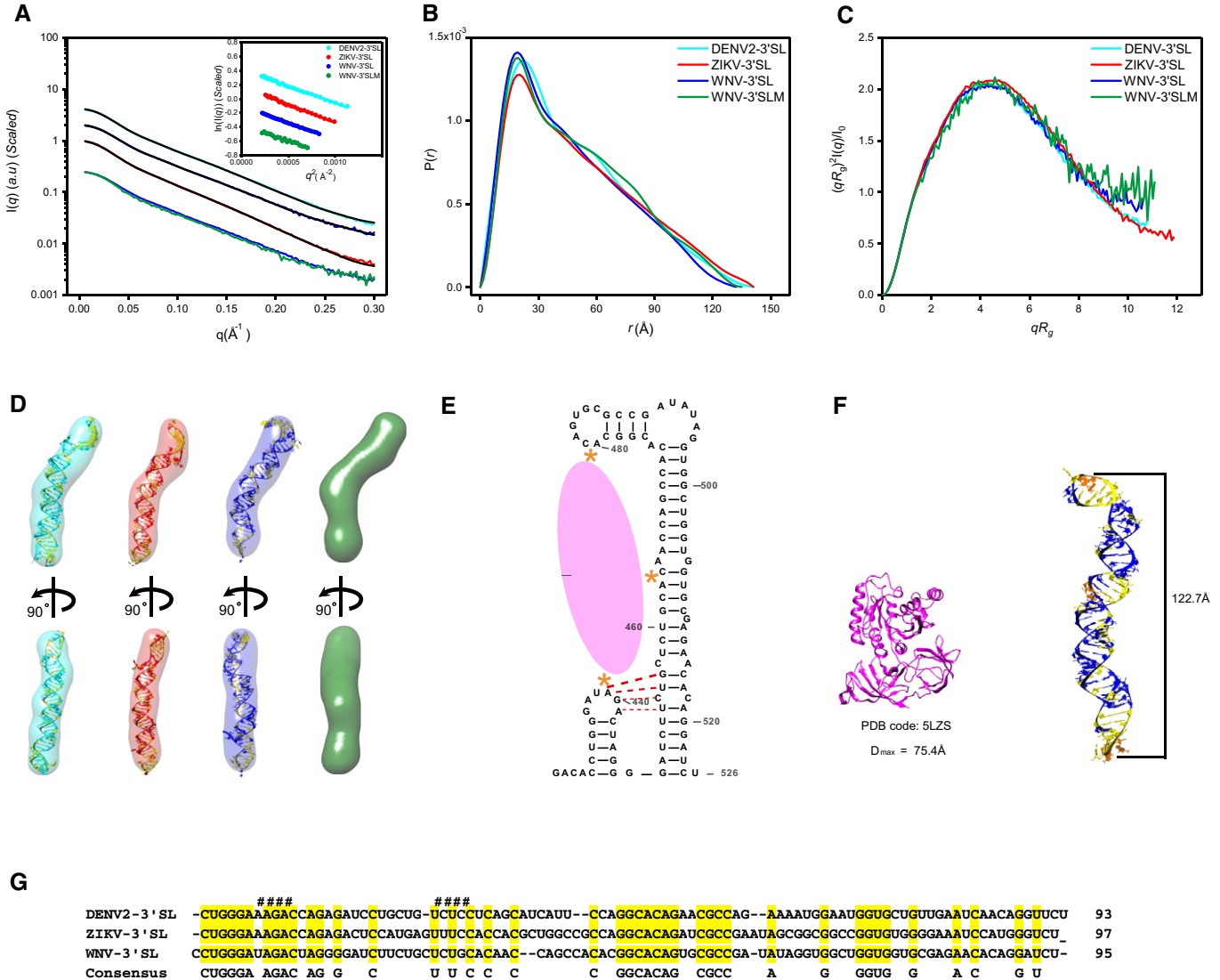

**Figure 7. SAXS analysis of the 3′SLs.**

A–C   The scattering profiles (A), the PDDFs (B), and the dimensionless Kratky plots (C) of 3′SLs from DENV2 (cyan), ZIKV (red), WNV (blue), and WNV-3′SLM ($^{438}$UAG$^{441}$A→$^{438}$GUC$^{441}$U) mutant (green). The inset in (A) shows the guinier regions of the respective scattering profiles with linear fit lines.

D   The *ab initio* shape envelopes of the 3′SLs of DENV2 (left), ZIKV (middle), and WNV (right), WNV-3′SLM (green). The atomic models from *de novo* structure modeling by Rosetta except for 3′SLM of WNV are superimposed onto the respective envelopes. The back-calculated scattering profiles of the *de novo* atomic models are fitted to the respective experimental scattering profiles in (A).

E   Illustration of the three binding sites (orange asterisks) of WNV-3′SL to eEF1A (pink ellipse), the potential pseudoknot is also indicated with red dash line.

F   The structure of ribosome-bound eEF1A (PDB: 5LZS) is compared with the envelope and atomic model of WNV-3′SL side by side.

G   Sequence alignment of DENV2-, ZIKV- and WNV-3′SL. The sequences involved in the potential pseudoknot between the sHP and long SL of WNV 3′SL were indicated with # sign at the top.

we employ the ensemble optimization method (EOM) approach [59]. In this approach, an initial random pool containing 10,000 coexisting conformers with different conformations that approximate (otherwise infinite) the possible conformational space are generated using the program Xplor-NIH. A genetic algorithm selection process then generates an optimized ensemble containing multiple conformers that best fit the experimental scattering curve [59]. Plots of the fitting chi-square between the experimental scattering curves and that calculated from the selected ensembles *versus* ensemble size ($N_e$) show that minimal ensembles with $N_e$

of 3, 3, and 3 for the DENV2-, ZIKV- and WNV sfRNAs, respectively, significantly improve the fitting, but further enlarging the ensemble size do not (Fig 8A–C), suggesting that the complete sfRNAs of DENV2, ZIKV, and WNV in solution are dynamic and can be best described as structural ensembles instead of one single structure. For each sfRNA, the selected ensemble has a narrow $R_g$ distribution in comparison with the initial random pool, and such a comparison suggests that in none of the sfRNAs are the individual subdomains free to articulate at random relative to their neighbors. The selected ensemble for sfRNA of DENV2 is skewed to

lower values of $R_g$ ($R_g$ (av) = 81 ± 20 Å; Appendix Table S3), suggesting that its structure is much more compact than those in the random pool. The selected ensembles for sfRNAs of ZIKV and WNV are skewed to higher values of $R_g$ ($R_g$ (av) = 89 ± 20 Å and $R_g$ (av) = 85 ± 15 Å, respectively; Appendix Table S3), indicating that the structures are more extended than an average random configuration (Fig 8A–C). The selected minimal structural ensembles consisting of three conformers for the complete sfRNAs are presented in Fig 8D–F, respectively. The $R_g$ values of the selected conformers in each minimal ensemble are listed in Appendix Table S3. In each ensemble, both compact and extended conformers are selected, suggesting that each sfRNA samples a large conformational space in solution.

## Discussion

In this work, on the basis of SAXS and computational modeling, we have outlined a general scheme to visualize the three-dimensional structures of a group of functionally important lncRNAs in solution, the sfRNAs from DENV2, ZIKV, and WNV that are important in virus replication, pathogenicity, and host immune evasion. We provide a complete and robust 3D structural models for the individual and combined subdomains as well as the accessible conformational spaces of the complete sfRNAs. The similarities and differences in the 3D structural models between different viruses shed mechanistic insights into their biological functions, which can aid in the development of functional hypotheses and experimental

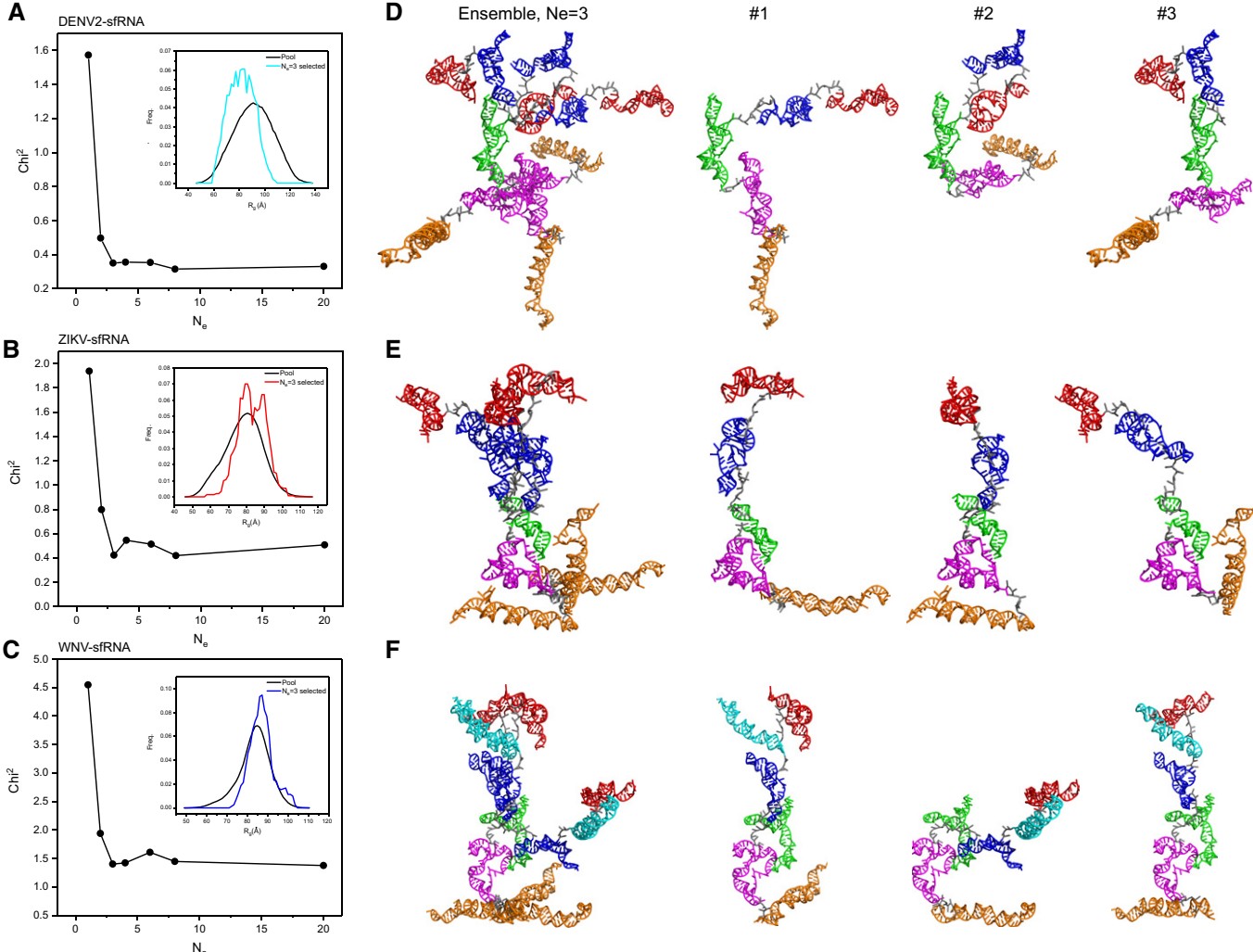

**Figure 8.  Structural ensembles of the complete sfRNAs.**

A–C   The fitting chi-square is plotted against the ensemble size for complete sfRNAs of DENV2 (A), ZIKV (B), and WNV (C), suggesting optimized minimal structural ensembles with 3, 3,3 conformers, respectively. The respective insets show the distribution of the minimal ensembles and the initial pools of 10,000 random conformers as a function of $Rg$.

D–F   The models of the selected minimal structural ensembles ($N_e$ = 3) that can reproduce the scattering curves of the complete sfRNAs of DENV2 (D), ZIKV (E), and WNV (F) are shown. In each panel, the left are the three individual models in each ensemble overlaid on top of the DB1 domains, the right are the three individual models in each ensemble displayed separately. The xrRNA1, xrRNA2, SL3 of WNV, ψ-DB1 and DB1, DB2, and 3′SL domains are colored in red, blue, cyan, green, magenta, and orange, respectively.

designs. Additionally, our work highlights the importance of the modular nature of lncRNAs like sfRNAs and SAXS can be a powerful technique to capture this type of structural organization.

Our SAXS analysis on xrRNA1-2s supports structural couplings between the duplicated xrRNA1 and xrRNA2. The Kratky plots show that xrRNA1-2s of DENV2, ZIKV, and WNV are well-folded in solution, implying reduced flexibility and potential structural coupling. The SAXS-derived shape structures reveal compact conformations and similar organization between xrRNA1 and xrRNA2 among the respective xrRNA1-2s (Fig 4D), suggesting that structural coupling between xrRNA1 and xrRNA2 may be common among the sfRNAs. Notably, the single-stranded linker between the two subdomains of ZIKV xrRNA1-2 exhibited decreased NMIA reactivities (Fig 1B), in support of local restricted accessibility imposed by structural coupling. The reduced flexibility and structural coupling may be a result of the topological constraints within the RNAs, which have been found to be major determinants of RNA structure and dynamics [60]. The structural coupling may also be mediated by unknown host factors, since all the functional couplings were observed *in vivo* and several host factors, such as Caprin 1, G3BP1/2 and USP10 were reported to bind to the xrRNA1 and xrRNA2 region of the 3′ UTR of DENV [55]. Due to the inherent limitation of SAXS data, the low-resolution shape envelopes and the atomic models of xrRNA1-2s do not suggest any defined residues that mediate the structural coupling; therefore, future high-resolution structure information is key to unveil the mechanism.

The secondary structure of the duplicated DB structures of DENV2 has been analyzed by SHAPE technique recently [13]. So far, no specific functions for the duplicated DB structures (DB12) have been identified, except that DB1 and DB2 may modulate viral replication in host-specific manner and could be the binding sites of specific interactions between the DENV2 sfRNA/3′UTR and the RNA helicase DDX6 [61], which is a multifunctional host protein that was implicated in translation regulation and the replication of several viruses, including hepatitis C viruses (HCV) and Dengue virus [62]. Disruption of the base pairing in the pseudoknots of DB1 and DB2 reduces the binding; therefore, pseudoknot formation is integral to the DBs' structure and function [61]. Previous secondary structure prediction suggests that DB12s of ZIKV and WNV form intertwined double pseudoknots significantly different from the sequential double pseudoknots in DENV2 DB12, but not experimentally verified. Our SHAPE and SAXS data support the formation of intertwined double pseudoknots in DB12s of ZIKV and WNV. Strikingly, our SAXS analysis reveals that the differences in double pseudoknots topology among DB12s from DENV2, ZIKV, and WNV result in quite different tertiary folding, while the SAXS-derived shape envelopes and atomic models of DB12s for ZIKV and WNV are compact, the DB12 structure of DENV2 is extended which suggests the two DB structures may act as independent functional units (Fig 5D). Of note, disruption of the original PK3s in ZIKV and WNV affects its structures *in vitro* significantly (Fig 5E), and the replication ability of ZIKV is also compromised upon PK3 mutations (Fig 6C–E). Interestingly, although the DB12 structures are quite different in ZIKV and DENV, the replication capacity of ZIKV was not significantly affected when the DB12 of ZIKV is exchanged with that of DENV2 (Fig 6C–E), suggesting that the biological function of these DBs in different flaviviruses might be complementary. Furthermore, whether and how the tertiary folding differences in

viral DB structures affect the binding properties to DDX6, and the function and pathogenic mechanism of the DB structures requires further investigation.

The shape envelopes and atomic models of 3′SLs of DENV2, ZIKV, and WNV reveal that they fold into similar extended rod-like, coaxially stacked helices, indicating conserved tertiary structures in solution. 3′SL is the most conserved structural elements across flaviviruses. Coaxial stacking is a well-known stabilizing factor in RNA structure. Previously, a putative pseudoknot formation between the small hairpin (sHP) and the long 3′ stem-loop (SL) in WNV was proposed [57], which will induce a bending in the shape envelope if formed. Sequence analysis shows that the corresponding sequences in the 3′ SLs of DENV2 and ZIKV are not conserved (Fig 7G); therefore, the potential interaction between the sHP and SL is not a conserved feature among flavivirus genomes. Our SAXS data and mutational analysis do not support the existence of such interaction or the interaction is too weak to be detected, consistent with the recent NMR studies [58]. Several host and viral proteins, including NS5, NS3, La autoantigen, and eEF1A, have been identified to interact with sfRNAs/3′UTR at the 3′SL site [21], but the structural basis of the interactions is unknown yet. RNA footprinting and filter binding assays identified one major and two minor binding sites for eEF1A on the 3′SL of WNV, a 5′-CACA-3′ sequence located on the 5′ side of the long stem-loop constitutes the major contact region, and the top loop of the long stem-loop and the sHP loop are the two minor eEF1A binding sites (Fig 7E) [63]. Recently, the high-resolution structure of mammalian eEF1A bound to ribosome was solved by cryo-EM [64], which consists of three subdomains that folds into a compact structure with a calculated $D_{max}$ of 75.4 Å (Fig 7F). In the SAXS-derived atomic model for 3′SLs, the top loop of the long stem-loop and the sHP loop are separated by an average of 122.7 Å (Fig 7F); therefore, the 3′SL may undergo a large conformational change upon eEF1A binding.

Our SAXS analysis on the complete sfRNAs indicates that sfRNAs are modular lncRNAs which sample large conformational spaces in solution therefore can be best described with structural ensembles instead of a single structure. The modularity of lncRNAs has been observed in many other cases, such as SRA [29], HOTAIR [28], and RepA [65], which encompass several independently folding modules separated with structures of high flexibility. Usually, the modular boundaries were derived on secondary structural domain map [13,29]. Based on previous secondary structure map, the sfRNA of DENV2 can be divided into five independently folding modules, xrRNA1, xrRNA2, DB1, DB2, and 3′ SL. Our SAXS analysis suggests the DB12s of ZIKV and WNV, respectively, should be considered as an independent folding module instead of two, highlighting the strength of SAXS as a powerful technique in capturing the modularity of lncRNA; therefore, DENV2-, ZIKV- and WNV sfRNAs consist of 5, 4, 4 modular subdomains. The modular nature of lncRNA is important in its function; for example, it enables sfRNAs to act as scaffolds, providing docking sites for different proteins in response to a variety of biological processes. sfRNAs/3′UTR have been identified to interact with many viral and host proteins including proteins involved in innate immunity (Trim25) [20], translation (eEF1A, PABP, La) [21], and mRNA metabolism (DDX6) [21], to facilitate their diverse functions by modulate several cellular pathways. Although which and how the subdomains are involved in specific proteins–RNA interactions remain largely unexplored, that the extended and open overall shape envelopes and the large

conformational spaces complete sfRNA sample in solution indicate that sfRNAs may readily adjust and expose the respective modular sites for binding to different proteins. For example, all the 3′SL domains are extruded out in structural ensembles of the complete sfRNAs (Fig 8D–F), therefore acting as an accessible dsRNA substrate for Dicer and/or other nucleases [27], but may undergo a dramatic conformational changes upon eEF1A binding.

Due to its large size and increased flexibility, structural characterization of lncRNA using conventional structural techniques including X-ray crystallography, NMR, and cryo-electron microscopy (cryo-EM) is extremely challenging, which, however, is essential to a complete understanding of the mechanisms and biological processes it involves in ref. [31]. The strong negative charges on the RNA backbone and the inherent flexibility of large RNA molecules often impede crystallization needed by X-ray crystallography. While NMR has proven to be a powerful technique in probing RNA structure and dynamics in solution, it usually limits to small RNAs (< 40 nucleotides). Cryo-EM has recently gained a quantum leap in the study of challenging biological systems, but there is usually a lower size limit (above ~200 kDa), and also RNA flexibility may prevent classification into a limited number of classes for single-particle reconstruction [66]. Therefore, current structural knowledge about lncRNA is mainly at the secondary structure level, and still very few lncRNA was characterized at the 3D level [31]. Our work highlights a powerful application of SAXS in combination with computational modeling in tertiary structural studies of large RNA. This general approach has been used to characterize the structures of other large RNAs, including the HIV-1 Rev response element [44], the T-box riboswitch core [67], the hepatitis C virus internal ribosome entry site RNA [47], and the HIV-1 5′ UTR RNAs [45]. Recently, a variety of robust programs that allow manual modeling, homology modeling, or *de novo* modeling of RNA 3D structures are available; however, "purely theoretical" structural predictions usually suffer from limited accuracy [68]. As shown above, SAXS data can be used to validate the secondary and tertiary structural models (such as the SL3 and 3′SL of WNV), or as experimental restraints in the structural computational modeling. With the availability of secondary structure information for more and more lncRNAs [31] and the development of RNA structure prediction algorithms [68], SAXS can be a powerful technique bridging the secondary and tertiary topological structures of large RNAs [34], therefore providing a direct visualization of the 3D structures of lncRNAs in solution.

# Materials and Methods

### RNA sample preparation

Plasmids coding an upstream T7 promoter and the DENV2-, WNV-, ZIKV sfRNA sequences, which correspond to nucleotides 10,301–10,723 of the Jamaica/N.1409 strain of a serotype 2 Dengue virus (GenBank M20588.1), nucleotides 11,520–11,042 of the West Nile virus strain HS101_08 isolated in South Africa (GenBank JN393308.1), and nucleotides 10,397–10,807 of the Zika virus strain SI-BKK01 isolated in Cambodia (GenBank, KY272987.1), were total gene synthesized and sequenced by Wuxi Qinglan Biotechnology Inc, Wuxi, China. Using these plasmids as templates, plasmids encoding

respective subdomains of sfRNAs were further constructed and confirmed by sequencing.

The double-stranded DNA fragment templates for *in vitro* RNA production were generated by PCR using an upstream forward primer targeted the plasmids and a downstream reverse primer specific to respective cDNAs. The RNAs were transcribed *in vitro* using T7 RNA polymerase and purified by preparative, non-denaturing polyacrylamide gel electrophoresis, and the target RNA bands were cut and passively eluted from gel slides into buffer containing 0.3 M NaOAc, 1 mM EDTA, pH 5.2 overnight at 4°C. The RNAs were further passed through the size exclusion chromatography (SEC) column or buffer exchanged extensively using Amicon Ultra Centrifugal Filter Devices (Millipore) to final buffer condition for SAXS. All RNA solutions were centrifuged at 16,000 *g* for 10 min and filtered through 0.22-µM syringe filter and diluted to final concentrations of 0.75–3 mg/ml immediately prior to SAXS measurement.

The locked nucleic acids (LNA) targeting ZIKV sfRNA were chemically synthesized and HPLC purified by Beijing SBS Genetech Co., Ltd. For preparation of ZIKV sfRNA-LNA complex sample, ZIKV sfRNA was mixed with LNA in a 1:10 molar ratio and incubated overnight at 4°C, further purified with SEC column, and then exchanged into SAXS buffer as above.

The sequences for the full-length and subdomain constructs of all sfRNAs, and the LNA sequence for this study are listed in Appendix Table S4.

### Dynamic light scattering

Dynamic light scattering studies were performed on a DynaPro NanoStar instrument equipped with a Temperature-Controlled MicroSampler (Wyatt Technology Corp., Santa Barbara, CA) at a laser wavelength of 660 nm, scattering angle of 90° in a 50-µl quartz cuvette at 25°C. Each measurement consisted of thirty 5-s acquisitions. All samples were centrifuged at 12,000 *g* for 10 min before measurements. To obtain the hydrodynamic radii ($R_h$) and percentage of polydispersity, the intensity autocorrelation functions were fitted with a non-negative least squares algorithm by *Dynamics 7.1.7.16* software (Wyatt Technology Corp., Santa Barbara, CA.).

### SHAPE analysis of the WNV and ZIKV sfRNAs

The selective 2′-hydroxyl acylation analyzed by primer extension (SHAPE) analysis was performed as described previously [39,69] with minor revisions. Briefly, 30 pmol of ZIKV or WNV sfRNA samples was diluted in 36 µl of 0.5×TE buffer (10 mM Tris–HCl, pH 8.0 and 1 mM EDTA for 1×). The diluted samples were denatured by heating at 95°C for 2 min and cooled immediately on ice, and 18 µl of 3.3×RNA folding buffer (333 mM HEPES, pH 8.0, 333 mM NaCl, and 20 mM MgCl₂) was added into the ice-cooled samples, which were then incubated at 37°C for 20 min for refolding. The samples were then divided into two 0.2-ml PCR tubes equally, and 3 µl of NMIA solution (Sigma-Aldrich, 130 mM in DMSO) was added in the SHAPE (+) reactions, whereas in the parallel SHAPE (−) reactions only 3 µl of DMSO was added. The SHAPE reactions were performed at 37°C for 45 min, and the modified/control RNA was purified by using RNA clean and concentrator-5 (Zymo research). Primer extension reactions as well as sequencing

reactions were performed as described previously [69] using the primers listed in Appendix Table S5. The denaturing PAGE capillary electrophoresis was performed by Sangon Biotech Co. SHAPE data were analyzed by the QuShape software [70]. Two independent tests were performed for the SHAPE analysis, and the mean values of normalized SHAPE reactivity were annotated on the structural models of WNV and ZIKV sfRNA, respectively. For the annotation of the WNV sfRNA, the 1–150 nt section of the SHAPE results was originated from the P-WNV3UTR-R2 reactions, and the SHAPE data of the 151–525 nt region were derived from the P-WNV3UTR-R1 reactions. Similar strategy was used for the annotation of the ZIKV sfRNA, in which the 1–140 nt section from the P-ZIKV3UTR-R2 reactions and the 141–412 nt section from the P-ZIKV3UTR-R1 reactions were combined to generate the SHAPE data.

### All-atom 3D atomic modeling

#### Comparative homology modeling
As the crystal structure of ZIKV-xrRNA1 is consistent with its solution structure, xrRNA1s and xrRNA2s are homologous in primary sequences, and similar in secondary structures, all-atom 3D models were built up for xrRNA1s and xrRNA2 from DENV2, ZIKV, WNV, and MVEV with ModeRNA [48], using the crystal structure of ZIKV-xrRNA1 (PDB: 5TPY) as a template.

#### De novo RNA structure prediction
Since no high-resolution structure is available for any of the flavivirus xrRNA3, xrRNA4, and 3′ SL, the SL3 of WNV, the program Rosetta [51], was used to generate ensembles of de novo all-atom 3D models for the SL3 of WNV, the xrRNA34, and the 3′ SLs of DENV2, ZIKV, and WNV, during which 2,000 structures were generated and energy minimized for each construct, respectively. Models in the ensembles were further screened against the experimental SAXS data, and the ones which have best fitting to the experimental SAXS data and can be nicely superimposed onto the bead models were chosen as the best representative all-atom 3D model for the respective constructs.

#### Rigid-body modeling
With the availability of atomic models for individual xrRNA1, xrRNA2, and SL3 of WNV, rigid-body modeling was carried out to construct atomic models for xrRNA12 of DENV2, ZIKV, and WNV using Xplor-NIH package [52], during which the respective individual subdomains were kept as rigid bodies, the linkers between were allowed to translate or rotate freely, and a simulated annealing algorithm was performed to optimize the best position and orientation of the individual domains against SAXS data.

### Small angle X-ray scattering

All the parameters for data collection and software employed for data analysis are summarized in Appendix Table S6. Details about the buffer conditions for respective RNAs in SAXS measurement are listed in Appendix Tables S1 and S2.

#### Data collection and processing
Small angle X-ray scattering measurements were carried out at room temperature at the beamline 12 ID-B of the Advanced Photon Source, Argonne National Laboratory or the beamline BL19U2 of the National Center for Protein Science Shanghai (NCPSS), and Shanghai Synchrotron Radiation Facility (SSRF). The scattered X-ray photons were recorded with a PILATUS 2 M detector (Dectris) at 12 ID-B and a PILATUS 100 k detector (Dectris) at BL19U2. The setups were adjusted to achieve scattering $q$ values of $0.005 < q < 0.89$ Å$^{-1}$ (12ID-B) or $0.009 < q < 0.415$ Å$^{-1}$ (BL19U2), where $q = (4\pi/\lambda)\sin\theta$, and $2\theta$ is the scattering angle. Thirty 2-dimensional images were recorded for each buffer or sample solution using a flow cell, with the exposure time of 0.5–2 s to minimize radiation damage and obtain good signal-to-noise ratio. No radiation damage was observed as confirmed by the absence of systematic signal changes in sequentially collected X-ray scattering images. The 2D images were reduced to one-dimensional scattering profiles using MATLAB (12ID-B) or BioXTAS Raw (BL19U2). Scattering profiles of the RNAs were calculated by subtracting the background buffer contribution from the sample-buffer profile using the program PRIMUS [71] following standard procedures [72]. Concentration series measurements (4- and 2-fold dilution and stock solution) for the same sample were carried out to remove the scattering contribution due to inter-particle interactions and to extrapolate the data to infinite dilution. The forward scattering intensity $I(0)$ and the radius of gyration ($R_g$) were calculated from the data of infinite dilution at low $q$ values in the range of $qR_g < 1.3$, using the Guinier approximation: $\ln I(q) \approx \ln(I(0)) - R_g^2 q^2/3$. These parameters were also estimated from the scattering profile with a broader $q$ range of 0.006–0.30 Å$^{-1}$ using the indirect Fourier transform method implemented in the program GNOM [73], along with the pair distance distribution function (PDDF), $p(r)$, and the maximum dimension of the protein, $D_{max}$. The parameter, $D_{max}$ (the upper end of distance $r$), was chosen so that the resulting PDDF has a short, near zero-value tail to avoid underestimation of the molecular dimension and consequent distortion in low-resolution structural reconstruction. The volume of correlation ($V_c$) was calculated using the program Scatter, and the molecular weights of solutes were calculated on a relative scale using the $R_g/V_c$ power law developed by Rambo and Tainer [38], independently of RNA concentration and with minimal user bias. The theoretical scattering intensity of the atomic structure model was calculated and fitted to the experimental scattering intensity using CRYSOL [74].

#### Ab initio shape reconstructions
Low-resolution bead models were built up with the program DAMMIN, which generate models represented by an ensemble of densely packed beads [43], using scattering data within the $q$ range of 0.006–0.30 Å$^{-1}$. Thirty-two independent runs were performed, and the resulting models were subjected to averaging by DAMAVER [75] and were superimposed by SUPCOMB [46] based on the normalized spatial discrepancy (NSD) criteria and were filtered using DAMFILT to generate the final model. NSD is a measure of quantitative similarity between sets of three-dimensional models, if two models systematically differ from each other, their NSD exceeds 1, for identical objects, it is 0.

#### Ensemble optimization method
One key step in EOM analysis is to generate a conformational pool containing a large set of models with different conformation to approximate (otherwise infinite) the conformational space. Starting

from the atomic models of the respective full-length sfRNAs, which were assembled with the available atomic models of individual subdomains as template in Rosetta, conformational pools were generated for the respective sfRNAs using the Xplor-NIH package, which follows a simulated annealing protocol driven by molecular dynamic simulation in torsion angle space that is subject to a target function comprising of bond length, bond angles, improper dihedral angles which specify chirality and planarity of functional groups, a quartic van der Waals repulsion, and $R_g$ terms to prevent atomic overlap, a multidimensional torsion angle database potential to improve backbone and sidechain conformation. During the simulation, subdomains of xrRNA1, xrRNA2, xrRNA3, xrRNA4, and 3′SL of DENV2 sfRNA; subdomains of xrRNA1, xrRNA2, xrRNA34, and 3′SL of ZIKV sfRNA; and subdomains of xrRNA1, SL3, xrRNA2, xrRNA34, and 3′SL of WNV sfRNA were treated as independent rigid bodies, and the linkers between were allowed to translate and/or rotate freely. To minimize the effect of the starting model conformation on sampling, two round calculations were carried out. A total of 120 models were generated by MD simulation in the first round, among which 20 models with high total energy and bond angle energy terms were picked out. The remaining 100 models were used as starting models for the next round calculation, which resulting in a total of 12,000 models. The same procedure as the first round was used to pick out those models with high total energy and bond angle energy terms, and a total of 10,000 models were left to assemble a conformational pool for respective sfRNAs. The respective conformational pool was used to search a minimal sub-ensemble that collectively reproduces the SAXS profile using the genetic algorithm GAJOE. The size of the minimal ensemble was varied from 1 to 20 (1, 2, 3, 4, 6, 8, 20) to test the effect of ensemble size on the quality of the fitting. Similar procedures were carried out for EOM analysis of xrRNA12 of DENV2, ZIKV, and WNV, and xrRNA34 of DENV2.

### Generation and evaluation of ZIKV mutants

#### Cell lines

Baby hamster kidney fibroblast cell line BHK-21 (ATCC CCL-10) was maintained at 37°C in 5% $CO_2$ in Dulbecco's modified Eagle's medium (DMEM, Gibco, Thermo Fisher Scientific), which contained 8% fetal bovine serum (FBS, Biowest) and 1% penicillin/streptomycin (PS, Thermo Fisher Scientific), respectively.

#### Plasmid construction

The corresponding nucleotide substitutions or insertions (Fig 6A and B) were introduced into the ZIKV infectious cDNA clone [76] (pFLZIKV) or Renilla luciferase reporter ZIKV replicon [56] using the Q5 site-directed mutagenesis kit (NEB) and verified by DNA sequencing. The sequence of the ZIKV 3′-UTR with the DBs replaced by the DENV2 DBs was chemically synthesized (Sangon Biotech) and sub-cloned into pFLZIKV by *Cla* I and *EcoR* I digestion. All mutants were confirmed by DNA sequencing. Rescue of ZIKV was performed as described previously [77]. The primers used for plasmid construction were listed in Appendix Table S7.

#### Immunofluorescence assay

BHK-21 cells were seeded onto 24-well plates, and transfection was performed using Lipofectamine 3000 reagent (Thermo Fisher Scientific) when the cells reached 60% confluency. The experiments were performed in triplicates. At indicated time points, the culture supernatants were collected and used to detect the viral RNA by qRT–PCR as described previously [78], the cells were fixed in acetone/methanol (V/V:3/7) at −20°C for 15 min, and ZIKV E protein expression was detected by IFA as described previously [69].

#### Replicon assay

BHK-21 cells were seeded onto 24-well plates and incubated at 37°C in 5% $CO_2$. Triplicate transfections were performed using Lipofectamine 3000 reagent (Thermo Fisher Scientific) at approximately 50% confluency, and a total of 200 ng replicon RNA was transfected into each well of BHK-21 cells. The cell lysates were collected at 6, 24, 36, 48, 60, and 72 h after transfection. Then Renilla luciferase activity was measured by the Renilla luciferase assay system (Promega) with a GloMax Discover multi-mode microplate reader (Promega).

### Structural illustration

All of the illustrations of atomic models were generated using the PyMOL Molecular Graphics System, version 1.3, Schrödinger, LLC. The bead models generated by DAMMIN were transformed into volumetric maps using Situs [79] and displayed using Chimera [80].

## Data availability

The experimental SAXS data and models for the respective RNAs have been deposited into the small angle scattering biological databank (SASBDB) (https://www.sasbdb.org/) with accession codes for DENV2 sfRNA: SASDG24, ZIKV sfRNA: SASDGZ3, WNV sfRNA: SASDG34, DENV2-xrRNA1: SASDGG3, ZIKV-xrRNA1: SASDGF3, WNV-xrRNA1: SASDGJ3, MVEV-xrRNA1: SASDGK3, DENV-xrRNA2: SASDGR3, ZIKV-xrRNA2: SASDGN3, WNV-xrRNA2: SASDGT3, MVEV-xrRNA2: SASDGU3, WNV-SL3: SASGQ3, DENV-xrRNA1-2: SASDGM3, ZIKV-xrRNA1-2: SASDGH3, WNV-xrRNA1-2: SASDGP3, DENV2-DB12: SASDGS3, ZIKV-DB12: SASDGV3, WNV-DB12: SASDGW3, DENV2-3′SL: SASDGL3, ZIKV-3′SL: SASDGX3, and WNV-3′SL: SASDGY3, respectively.

**Expanded View** for this article is available online.

### Acknowledgements

This research used resources of the Advanced Photon Source, a U.S. Department of Energy (DOE) Office of Science User Facility operated for the DOE Office of Science by Argonne National Laboratory under Contract No. DE-AC02-06CH11357. We thank Dr. Xiaobing Zuo at the beamline 12-ID-B, Advanced Photon Source, Argonne National Laboratory and the staffs of beamline BL19U2 at National Facility for Protein Science Shanghai (NFPS), and Shanghai Synchrotron Radiation Facility, Shanghai, People's Republic of China for assistance during data collection. This work was supported by grants from the National Key Research and Development Project of China (2016YFA0500700), the China Youth 1000-Talent Program of the State Council of China, Tsinghua University Initiative Scientific Research Program, the Beijing Advanced Innovation Center for Structural Biology, the Tsinghua-Peking Joint Center for Life Sciences to X.F., and the National Key Research and Development Project of China (2016YFD0500304, 2018ZX09711003), and the National Natural Science Foundation of China (No. 81522025, 81621005, 31770190) to C.F.Q.

## Author contributions

XF conceived the project, designed the experiments, and wrote the manuscript. YZ, YZ performed and interpreted the majority of the experiments. Z-YL performed the SHAPE experiments and analyzed the data. M-LC performed the *in vivo* assay based on replicon and infectious clone system. JM and YW prepared some of the RNA samples. C-FQ supervised the SHAPE experiments and *in vivo* assay. Z-YL and C-FQ contributed on experiment design, data interpretation, and manuscript editing.

## Conflict of interest

The authors declare that they have no conflict of interest.

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
