## [Review Process File · EMBO Reports]

Long non-coding subgenomic flavivirus RNAs have extended 3D structures and are flexible in solution

Yupeng Zhang, Yikan Zhang, Zhong-Yu Liu, Meng-Li Cheng, Junfeng Ma, Yan Wang, Cheng-Feng Qin, Xianyang Fang

Review timeline:

Submission date:	6 September 2018
Editorial Decision:	10 October 2018
Revision received:	29 June 2019
Editorial Decision:	6 August 2019
Revision received:	14 August 2019
Accepted:	21 August 2019

Editor: Achim Breiling

Transaction Report:

1st Editorial Decision

10 October 2018

Thank you for the submission of your research manuscript to EMBO reports. We have now received reports from the three referees that were asked to evaluate your study, which can be found at the end of this email.

As you will see, all referees think the manuscript is of interest, but requires a major revision to allow publication in EMBO reports. All three referees have a number of concerns and/or suggestions to improve the manuscript, which we ask you to address in a revised manuscript. As the reports are below, I will not detail them here. However, it will be a major prerequisite to consider publication here that the major conclusions of the manuscript are tested experimentally (as indicated by referee #1) and validated (as indicated by referee #3) *in vivo*, and are placed back into a biological relevant context. Further, as indicated by referees #1 and #2, the paper needs an extensive revision regarding grammar and language, and needs to be proofread by a native speaker.

Looking at the referee comments, a significant revision is required before publication of your manuscript can be considered, and I would also understand your decision if you chose to rather seek rapid publication elsewhere at this stage. However, given the constructive referee comments, I would like to invite you to revise your manuscript with the understanding that all referee concerns must be addressed in the revised manuscript and/or in a detailed point-by-point response.

Acceptance of your manuscript will depend on a positive outcome of a second round of review. It is EMBO reports policy to allow a single round of revision only and acceptance or rejection of the manuscript will therefore depend on the completeness of your responses included in the next, final version of the manuscript.

Revised manuscripts should be submitted within three months of a request for revision; they will otherwise be treated as new submissions. Please contact us if a 3-months time frame is not sufficient for the revisions so that we can discuss the revisions further.

Supplementary/additional data: The Expanded View format, which will be displayed in the main HTML of the paper in a collapsible format, has replaced the Supplementary information. You can submit up to 5 images as Expanded View. Please follow the nomenclature Figure EV1, Figure EV2 etc. The figure legend for these should be included in the main manuscript document file in a section called Expanded View Figure Legends after the main Figure Legends section. Additional Supplementary material should be supplied as a single pdf labeled Appendix. The Appendix includes a table of content on the first page, all figures and their legends. Please follow the nomenclature Appendix Figure Sx throughout the text and also label the figures according to this nomenclature.

For more details please refer to our guide to authors:
<http://embor.embopress.org/authorguide#manuscriptpreparation>

Important: All materials and methods should be included in the main manuscript file.

See also our guide for figure preparation:
http://www.embopress.org/sites/default/files/EMBOPress_Figure_Guidelines_061115.pdf

Regarding data quantification and statistics, can you please specify, where applicable, the number "n" for how many independent experiments (biological replicates) were performed, the bars and error bars (e.g. SEM, SD) and the test used to calculate p-values in the respective figure legends. Please provide statistical testing where applicable. See:
<http://embor.embopress.org/authorguide#statisticalanalysis>

We now strongly encourage the publication of original source data with the aim of making primary data more accessible and transparent to the reader. The source data will be published in a separate source data file online along with the accepted manuscript and will be linked to the relevant figure. If you would like to use this opportunity, please submit the source data (for example scans of entire gels or blots, data points of graphs in an excel sheet, additional images, etc.) of your key experiments together with the revised manuscript. Please include size markers for scans of entire gels, label the scans with figure and panel number, and send one PDF file per figure.

- a complete author checklist, which you can download from our author guidelines (<http://embor.embopress.org/authorguide#revision>). Please insert page numbers in the checklist to indicate where the requested information can be found.
- a letter detailing your responses to the referee comments in Word format (.doc)
- a Microsoft Word file (.doc) of the revised manuscript text
- editable TIFF or EPS-formatted single figure files in high resolution (for main figures and EV figures)

Please also note that we now mandate that all corresponding authors list an ORCID digital identifier that is linked to their EMBO reports account!

I look forward to seeing a revised version of your manuscript when it is ready. Please let me know if you have questions or comments regarding the revision.

REFeree REPORTS

Referee #1:

The report from Zhang et al is interesting and timely since the study of sRNAs, a non-coding RNAs, produced during flavivirus infection is one of the most exciting aspects of this field. I can only comment on the general RNA biology and virology and not the structural biology - therefore on technical aspects of the use and interpretation of SAXS data I defer to structural biology reviewers.

In my opinion the manuscript is a strong contribution to the field and of high general interest, but could be further enhanced by the following:

- 1) The manuscript should be extensively revised to correct issues of language - the first paragraph is a good example of this (e.g., the tense used is changed inappropriately). I strongly suggest that it be subjected to editing by a professional service. Equally, and also dealing with form, the review of the virology literature was at times wrong (e.g., flaviviruses cause tens of thousands of death a year and hundreds of millions of infections....). I believe that these errors reflect a language problem, but the authors should strive to be accurate in their summary of the literature. The text that addressed the previous work on sfRNA structure appears better. That said terms like 'confounding' and 'exoribonuclease' seemed directly borrowed from other texts.
2. The manuscript would be improved if the authors can experimentally test some of the conclusions of the manuscript - examples below:
 - a) The idea that the sequence between SLI/SLII and DB1/DB2 is flexible and may mediate the different conformations predicted. Can the authors bind a short antisense oligonucleotide or LNA to this region and change the overall structure observed (or variability of structure in SAXS experiment with full length molecule). In fact this sequence is not SHAPE reactive in DENV2 - this is puzzling.
 - b) The finding that the data are not consistent with the predicted pseudoknot in the 3' terminal structure is of interest - it would be useful to make mutations that significantly enhance this non-conserved feature and those that significantly cripple it to see if this alters the SAXS results.
- 3) The authors should comment on the high SHAPE reactivity of WNV sfRNA nts 120 and 124 - should this terminal loop be re-drawn?

Referee #2:

This is an impressive paper that prevents a comparative structural analysis of an important noncoding region of flaviviral genomes. The manuscript is significant on many levels. It presents an innovative combination of SAXS with a suite of robust new molecular modeling programs to generate the first clear structural view of sfRNAs, which play a key role in regulating viral replication. The paper and the valuable methodological advances therein are very timely, as the need for structural information on large RNAs is increasing and the tools have historically been limited. Although much attention is devoted to cryo-EM studies of macromolecular structure, SAXS has a valuable place in the arsenal because the molecules are truly in solution and it can be used to visualize RNA structures that are not uniformly compact, but composed of modular compact units that are separated by structures with high degrees of freedom. The work is scientifically important because it underscores the importance of appreciating the modular nature of large, folded RNA molecules and it highlights the importance of techniques that can capture this type of molecular organization. In terms of virology, it provides concrete information on significant differences in sfRNA structure between different viruses, suggesting that they do not all function in the same way. Because the approaches are so novel and also because there has been such an emphasis on cryo-EM over SAXS in recent years, this review will include some detail on strengths of the paper, and it will not only emphasize weaknesses.

Strengths:

1. Page 7: Attention to the quality of the samples is commendable and unusually rigorous. The RNAs were generated by native purification, but they were not assumed to be folded correctly. Rather, they were carefully examined by SAXS and other methods to evaluate homogeneity. This reviewer particularly liked the application of SAXS for monitoring the Mg²⁺ dependence of folding (Appendix Fig. 1), which is a much more informative way to evaluate transitions to a compact conformation.
- 2, Fig. 3: The results and the modeling for the xrRNA1 and 2 tandem domains is very impressive, clearly showing differences between the different viruses.
3. page 14, bottom paragraph. The approach for building up atomic models was excellent and

utilizes the powerful new techniques available for including sparse data along with subdomain models. Modeling RNA architecture is challenging, and the authors provide a clear prescription about how to proceed that will be helpful for many different types of projects and input data.

4. page 16: "the sequences involved in PK3 and PK4 formation in ZIKV and WNV DB12s are in proximity and locate in the same side after DB2, therefore, facilitate closed and compact conformations. The importance of formation of the PK3 was further evaluated by mutagenesis analysis". The fact that the disposition of the PK3 region in ZIKV and WNV causes such as profound alteration of molecular shape is fascinating, and gives insights into the evolution of RNA tertiary structures. The results are enhanced by the mutational analysis done in parallel.

Weaknesses and suggestions:

1. Page 8: The language is not clear, and still not clear when looking at figure. - "Interestingly, both the sequences predicted to form PK3 and PK4 with ψ -DB1 and DB2 in ZIKV sfRNA, DB1 and DB2 in WNV sfRNA, respectively, locate in the same side of sequences after DB2 (Fig 1B, C). In contrast, the sequences involved in PK3 and PK4 formation in DB1 and DB2 in DENV2 sfRNA are located before and after DB2 structure". Here and throughout the paper, the manuscript could use more careful editing for language and spelling. For example, later in the paper the term "medicated" was used in place of "mediated", and there are other examples.

2. Page 12 top paragraph: Does it make sense and is it feasible to examine the SAXS envelope of xrRNA1 and xrRNA2 in isolation, as they are only considered in tandem here. Given that the calculation for the intact model is highly constrained, would it not be desirable to have envelopes for individual domains? This reviewer was left feeling that more SAXS experiments on the individual RNA subdomains would have resulted in better models. So I was curious why this was not a more general part of the approach.

3. More of the figures in the Appendix need to be brought into the main text. For example, representative figures should be taken from Appendix fig 4 and put into main manuscript.

4. The paper would benefit from a final paragraph that summarizes the significance of the paper in terms of virology, in terms of lncRNA structure, and in terms of methodology. The Discussion gets into the weeds a bit without every really wrapping up what is interesting about this paper.

Referee #3:

In this work, Zhang et al construct simplified RNAs corresponding to parts of the sfRNAs of several flaviviruses and analyze the structures of these RNAs, primarily by SAXS. The work is timely in that interest in these viruses and sfRNAs is high, but it is not clear that much has really been learned about the structure.

Most of the models could probably have been generated without SAXS data and the major conclusion of the work, that these RNAs form conformational ensembles (Fig. 5) is not surprising.

Major concerns:

The writing and conclusions are diffuse with potentially interesting modeling, but no real validation. In some cases models were reported to be different from crystal structures (MVEV, p. 13), but it was not made clear how.

Multiple modeling programs (ModeRNA, Rosetta, Xplor-NIH) were used but no analysis of which is better or of the quality of any of these models.

WNV SLI-II RNA reported as having a different tertiary organization than DENV2 and ZIKV, but this is an obvious likelihood as the secondary structure contains an extra stem loop (compare Figs. 1 and 3).

I do not understand the statement, "unexpectedly, the shape envelopes for DB12s of ZIKV and

WNV are compact and closed (Fig 4D)" Why is this unexpected? There is an intertwined double pseudoknot in both structures. This is perhaps the most interesting conclusion of the paper, but seems to follow directly from the (known) secondary structure.

Minor concern:

It is not clear what mutant was made in Fig. 4E.

1st Revision - authors' response

29 June 2019

Response to referee 1

Referee #1:

The report from Zhang et al is interesting and timely since the study of sfRNAs, a non-coding RNAs, produced during flavivirus infection is one of the most exciting aspects of this field. I can only comment on the general RNA biology and virology and not the structural biology - therefore on technical aspects of the use and interpretation of SAXS data I defer to structural biology reviewers.

In my opinion the manuscript is a strong contribution to the field and of high general interest, but could be further enhanced by the following:

OR2: We appreciate the positive comment from Referee 1 with regard to the scientific significance about the work.

1) The manuscript should be extensively revised to correct issues of language - the first paragraph is a good example of this (e.g., the tense used is changed inappropriately). I strongly suggest that it be subjected to editing by a professional service. Equally, and also dealing with form, the review of the virology literature was at times wrong (e.g., flaviviruses cause tens of thousands of death a year and hundreds of millions of infections....). I believe that these errors reflect a language problem, but the authors should strive to be accurate in their summary of the literature. The text that addressed the previous work on sfRNA structure appears better. That said terms like 'confounding' an exoribonuclease seemed directly borrowed from other texts.

OR3: We appreciate and fully agree with Referee 1's comments. The language has been carefully reviewed and the inappropriate tense and words in the text have been corrected.

2. The manuscript would be improved if the authors can experimentally test some of the conclusions of the manuscript - examples below:

a) The idea that the sequence between SLI/SLII and DB1/DB2 is flexible and may mediate the different conformations predicted. Can the authors bind a short antisense oligonucleotide or LNA to this region and change the overall structure observed (or variability of structure in SAXS experiment with full length molecule). In fact this sequence is not SHAPE reactive in DENV2 - this is puzzling.

OR4: We appreciate the referee's suggestion. (1) A complementary LNA is synthesized to target the sequence between SLI/SLII and DB1/DB2 of ZIKV-sfRNA. As the referee expects, LNA binding to this region cause significant overall structural changes in ZIKV-sfRNA, the PAGE gel, the scattering curves and the PDDFs, the shape envelopes are shown in EV Fig 5G-J. (2) We also design complementary LNAs to target the linkers between SLI/SLII and DB1/DB2 in DENV- and WNV- sfRNAs, but no obvious binding is observed, we therefore don't show the data. The corresponding linker in DENV-sfRNA is AU-rich, the corresponding linker in WNV-sfRNA is relatively short, which may result in poor binding of LNA to these regions. (3) We guess referee 1 is confused with Fig 1A, which shows the secondary structure of DENV2-sfRNA, but with no SHAPE reactivities annotated. Actually, we didn't do SHAPE analysis for DENV2-sfRNA in our work, but which has been done by another group (Elife 2014, see reference 13), so no SHAPE reactivity data is mapped to the secondary structure of DENV-sfRNA in Fig 1A. In Elife 2014 (reference 13), the sequence between SLI/SLII and DB1/DB2 in DENV2 3'-UTR is shape reactive.

b) The finding that the data are not consistent with the predicted pseudoknot in the 3' terminal

structure is of interest - it would be useful to make mutations that significantly enhance this non-conserved feature and those that significantly cripple it to see if this alters the SAXS results.

OR5: As the referee suggests, we make a mutant of WNV 3'SL (WNV-3'SLM: ⁴³⁸UAG⁴⁴¹A→⁴³⁸GUC⁴⁴¹U), which will prevent any pseudoknot formation. We characterize the mutant with SAXS, and no significant conformational changes can be observed, we include this result in Fig. 7D.

3) The authors should comment on the high SHAPE reactivity of WNV sfRNA nts 120 and 124 - should this terminal loop be re-drawn?

OR6: The referee is correct. We correct the secondary structure of WNV sfRNA at nucleotide 120 and 124, the correction has been made in Fig 1C.

Response to referee 2

Referee #2:

This is an impressive paper that prevents a comparative structural analysis of an important noncoding region of flaviviral genomes. The manuscript is significant on many levels. It presents an innovative combination of SAXS with a suite of robust new molecular modeling programs to generate the first clear structural view of sfRNAs, which play a key role in regulating viral replication. The paper and the valuable methodological advances therein are very timely, as the need for structural information on large RNAs is increasing and the tools have historically been limited. Although much attention is devoted to cryo-EM studies of macromolecular structure, SAXS has a valuable place in the arsenal because the molecules are truly in solution and it can be used to visualize RNA structures that are not uniformly compact, but composed of modular compact units that are separated by structures with high degrees of freedom. The work is scientifically important because it underscores the importance of appreciating the modular nature of large, folded RNA molecules and it highlights the importance of techniques that can capture this type of molecular organization. In terms of virology, it provides concrete information on significant differences in sfRNA structure between different viruses, suggesting that they do not all function in the same way. Because the approaches are so novel and also because there has been such an emphasis on cryo-EM over SAXS in recent years, this review will include some detail on strengths of the paper, and it will not only emphasize weaknesses.

Strengths:

1. Page 7: Attention to the quality of the samples is commendable and unusually rigorous. The RNAs were generated by native purification, but they were not assumed to be folded correctly. Rather, they were carefully examined by SAXS and other methods to evaluate homogeneity. This reviewer particularly liked the application of SAXS for monitoring the Mg²⁺ dependence of folding (Appendix Fig. 1), which is a much more informative way to evaluate transitions to a compact conformation.

2, Fig. 3: The results and the modeling for the xrRNA1 and 2 tandem domains is very impressive, clearly showing differences between the different viruses.

3. page 14, bottom paragraph. The approach for building up atomic models was excellent and utilizes the powerful new techniques available for including sparse data along with subdomain models. Modeling RNA architecture is challenging, and the authors provide a clear prescription about how to proceed that will be helpful for many different types of projects and input data.

4. page 16: "the sequences involved in PK3 and PK4 formation in ZIKV and WNV DB12s are in proximity and locate in the same side after DB2, therefore, facilitate closed and compact conformations. The importance of formation of the PK3 was further evaluated by mutagenesis analysis". The fact that the disposition of the PK3 region in ZIKV and WNV causes such a profound alteration of molecular shape is fascinating, and gives insights into the evolution of RNA tertiary structures. The results are enhanced by the mutational analysis done in parallel.

OR7: We thank the referee for his/her positive and insightful comments on our work. The comments actually help us revise the manuscript, such as reminding us what points we should emphasize. Guided by referee 2's comments, we rewrite some parts of the manuscript.

Weaknesses and suggestions:

1. Page 8: The language is not clear, and still not clear when looking at figure. - "Interestingly, both the sequences predicted to form PK3 and PK4 with ψ -DB1 and DB2 in ZIKV sfRNA, DB1 and DB2 in WNV sfRNA, respectively, locate in the same side of sequences after DB2 (Fig 1B, C). In contrast, the sequences involved in PK3 and PK4 formation in DB1 and DB2 in DENV2 sfRNA are located before and after DB2 structure". Here and throughout the paper, the manuscript could use more careful editing for language and spelling. For example, later in the paper the term "medicated" was used in place of "mediated", and there are other examples.

OR8: We thank the referee's criticism. We have changed our description about the topological organization of DB12s in the text (page 8-9) and corrected the term "medicated" as the referee points out. We also carefully reviewed the text and correct those mistakes in language and spelling.

2. Page 12 top paragraph: Does it make sense and is it feasible to examine the SAXS envelope of xrRNA1 and xrRNA2 in isolation, as they are only considered in tandem here. Given that the calculation for the intact model is highly constrained, would it not be desirable to have envelopes for individual domains? This reviewer was left feeling that more SAXS experiments on the individual RNA subdomains would have resulted in better models. So I was curious why this was not a more general part of the approach.

OR9: We thank the referee's suggestions. We actually have studied the individual xrRNA1s and xrRNA2s from DENV1, ZIKV, WNV and MVEV by SAXS, the related data was put in the Appendix. We now bring the data back to the main text with a new subsection (see page 11-13 in revised manuscript)

3. More of the figures in the Appendix need to be brought into the main text. For example, representative figures should be taken from Appendix fig 4 and put into main manuscript.

OR10: We thank the referee's suggestions. We have done this as suggested.

4. The paper would benefit from a final paragraph that summarizes the significance of the paper in terms of virology, in terms of lncRNA structure, and in terms of methodology. The Discussion gets into the weeds a bit without every really wrapping up what is interesting about this paper.

OR11: We thank the referee's suggestions. We have rewritten the discussion part. We delete the discussion about the potential interaction between xrRNA1 and SL3 in WNV-sfRNA, we rewrite the paragraph about the modular feature of lncRNA structure, a new paragraph about the potential application of SAXS in lncRNA structure study.

Response to referee 3**Referee #3:**

(1) In this work, Zhang et al construct simplified RNAs corresponding to parts of the sfRNAs of several flaviviruses and analyze the structures of these RNAs, primarily by SAXS. The work is timely in that interest in these viruses and sfRNAs is high, but it is not clear that much has really been learned about the structure.

OR12: We appreciate the referee's positive comment on our work.

(2) Most of the models could probably have been generated without SAXS data and the major conclusion of the work, that these RNAs form conformational ensembles (Fig. 5) is not surprising.

OR13: (1) We agree with the referee that most of the models could have been generated without SAXS based on pure computational methods. We also generate models with computational methods, but the accuracy of the models can be further evaluated and refined with experimental SAXS data, which should improve the models. For example, SAXS data doesn't support the pseudoknot interaction in the 3'SL which has been predicted before, but support the double pseudoknot formation in DB12s from ZIKV and WNV, SAXS data also validate the solution structure of individual xrRNA1s and xrRNA2s as the crystal structure of ZIKV-xrRNA1, not the crystal structure of xrRNA2 from MVEV; (2) We agree with the referee that these sfRNAs form conformational ensemble in solution is not surprising due to its increased flexibility, but there is only very few method that can describe the flexibility of large

RNAs. One of the strengths in SAXS is its application in flexibility analysis, which have been widely used for protein study. Our ensemble analysis on sRNAs is the first of a few applications of SAXS in flexibility analysis for large RNAs.

Major concerns:

(3) *The writing and conclusions are diffuse with potentially interesting modeling, but no real validation. In some cases models were reported to be different from crystal structures (MVEV, p. 13), but it was not made clear how.*

OR14: (1) We appreciate the referee's criticism. We have generated several mutants to validate the respective models by SAXS and included the results in revised manuscript (Fig 5C&E, Fig 7C&D, Fig EV5). (2) To make the differences in crystal structures from ZIKV and MVEV clear, we add the structural comparison in Fig 3A-B.

(4) *Multiple modeling programs (ModeRNA, Rosetta, Xplor-NIH) were used but no analysis of which is better or of the quality of any of these models.*

OR15: We appreciate the referee's criticism. We should make it more clear in the text that different modeling programs are used in different circumstances and for different purposes. When high resolution structures are available (such as xrRNA1 from ZIKV and xrRNA2 from MVEV), ModeRNA can be used for homology modeling as long as the primary and secondary structures are conserved. If there is no homology model available (such as DB12s, 3'SLs), the program Rosetta can be used to build up de novo atomic models, but which can be further validated and refined with experimental restraints like SAXS. Xplor-NIH is a program for rigid-body modeling of multi-domain RNAs against experimental restraints such as SAXS, given that the atomic models of subdomains are available.

As there are no more than 1,400 RNA structures in PDB which has a total entry more than 150,000, and RNA structure determination by experimental methods is challenging, our structural modeling in combination with SAXS data can provide reasonable models that are consistent with the experimental data, herein adding a further layer of restraints in the modeling to improve the accuracy of the models. It's therefore hard to compare the quality of these models from different programs.

(5) *WNV SLI-II RNA reported as having a different tertiary organization than DENV2 and ZIKV, but this is an obvious likelihood as the secondary structure contains an extra stem loop (compare Figs. 1 and 3).*

OR16: We appreciate the referee's criticism. We change the text in description about WNV SLI-II RNA accordingly.

(6) *I do not understand the statement, "unexpectedly, the shape envelopes for DB12s of ZIKV and WNV are compact and closed (Fig 4D)" Why is this unexpected? There is an intertwined double pseudoknot in both structures. This is perhaps the most interesting conclusion of the paper, but seems to follow directly from the (known) secondary structure.*

OR17: We appreciate the referee's criticism. (1) We delete the word "unexpectedly" in the text. This part has been rewritten to better describe the findings. (2) We would like to explain the strength of SAXS and computation in DB12 3D structure study. There are many cases in the literature that secondary structure models from computational or SHAPE analysis may not be correct, so validation of secondary structure by 3D structural techniques experimentally is important. Our SAXS analysis clearly validate the double pseudoknot formation in DB12s from ZIKV and WNV, which also show significant differences in 3D folding of DB12s from different viruses. (3) We have also evaluated the biological significance of ZIKV DB12 in virus replication and translation using in vivo assay, which was reported in a new subsection in the text (page 17-19) and a new Fig 6. Although we can't answer its mechanism in the work, it invokes new proposal for future study.

Minor concern:

It is not clear what mutant was made in Fig. 4E.

OR18: We appreciate the referee's criticism. In the revised version, we explain the mutant in the text (page 17) and Appendix Table S4. We also include the shape envelopes of the mutants in Fig 5C, E.

Thank you for the submission of your revised manuscript to our editorial offices. We have now received the reports from the two referees that were asked to re-evaluate your study, you will find below. As you will see, both referees now support the publication of your study in EMBO reports.

Before we can proceed with formal acceptance, I have these editorial requests:

- Could you provide a more comprehensive and active title, that mentions the major outcome of the study (with not more than 100 characters including spaces)
- Please add up to five key words to the title page.
- Please upload also the EV figures as individual production quality figure files as .eps, .tif, or .jpg (one file per figure). Please upload these as separate, individual files. Please then remove the EV Figures from the Appendix.
- Please remove the sentence "Expanded View for this paper is available Online" from the manuscript text.

Please have your manuscript carefully proofread by a native speaker. There are still several typos present.

- We require that primary datasets produced in this study and computational models are deposited in an appropriate public database. See:
<http://www.embopress.org/page/journal/14693178/authorguide#datadeposition>

The accession numbers and database should be listed in a formal "Data Availability" section (placed after Materials & Methods) that follows the model below. Please do that for your manuscript. Please note that the Data Availability Section is restricted to new primary data that are part of this study.

Data availability

- Please indicate the nature of replicates (n) in all the figure legends (e.g. biological or technical).
- Finally, please find attached a word file of the manuscript text (provided by our publisher) with changes we ask you to include in your final manuscript text, and queries, we ask you to address. Please provide your final manuscript file with track changes, in order that we can see the modifications done.

In addition I would need from you:

- a short, two-sentence summary of the manuscript
- two to three bullet points highlighting the key findings of your study
- a schematic summary figure (in jpeg or tiff format with the exact width of 550 pixels and a height of not more than 400 pixels) that can be used as a visual synopsis on our website.

REFEREE REPORTS

Referee #1:

The authors appropriately dealt with my comments.

Referee #2:

The manuscript is suitable for publication in EMBO reports without further revision.

2nd Revision - authors' response

14 August 2019

The authors performed all minor editorial changes.

Corresponding Author Name:	Xianyang Fang
Manuscript Number:	EMBOR-2019-48705